# MELODI: EXPLORING MEMORY COMPRESSION FOR LONG CONTEXTS

**Yinpeng Chen**          **DeLesley Hutchins**          **Aren Jansen**
**Andrey Zhmoginov**          **David Racz**          **Jesper Andersen**
Google DeepMind
{yinpengc,delesley,arenjansen,azhmogin,dracz,jespera}@google.com

## ABSTRACT

We present MELODI, a novel memory architecture designed to efficiently process long documents using short context windows. The key principle behind MELODI is to represent short-term and long-term memory as a hierarchical compression scheme across both transformer layers and context windows. Specifically, the short-term memory is achieved through recurrent compression of context windows across multiple layers, ensuring smooth transitions between windows. In contrast, the long-term memory performs further compression within a single middle layer and aggregates information across context windows, effectively consolidating crucial information from the entire history. Compared to a strong baseline - the Memorizing Transformer employing dense attention over a large long-term memory (64K key-value pairs) - our method demonstrates superior performance on various long-context datasets while remarkably reducing the memory footprint by a factor of 8.

## 1 INTRODUCTION

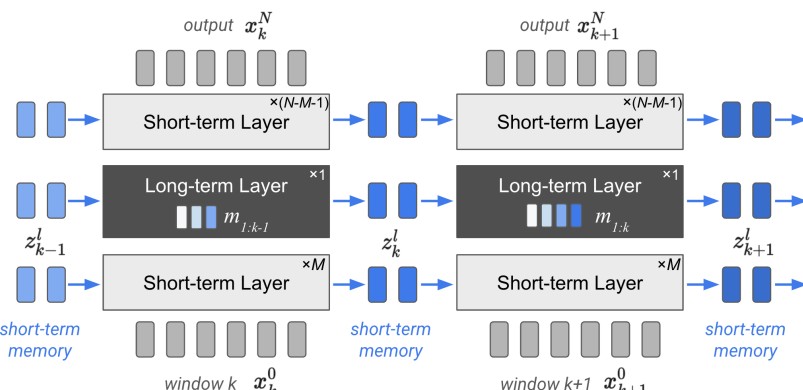

Figure 1: **Overview of MELODI.** MELODI employs a hierarchical memory representation, incorporating both short-term and long-term compression mechanisms, integrated with a transformer-based language model. It utilizes a stack of short-term layers to recurrently compress each context window $x_k^0$ into short-term memory tokens $\{z_k^l\}$, and inserts a long-term layer to store compressed key-value pairs within a long-term memory $m_{1:k}$. Both short-term and long-term layers leverage modified transformer blocks. In this illustration, we assume a total of $N$ layers, with $M$ short-term layers preceding 1 long-term layer and $N - M - 1$ short-term layers following it.

Long-context language models, exemplified by Gemini (Gemini-Team, 2024) and GPT (OpenAI, 2024), showcase remarkable capabilities across diverse modalities (e.g., text, images, audio, code, video) and seamlessly integrate various machine learning techniques, including many-shot in-context learning (Agarwal et al., 2024), chain-of-thought prompting (Wei et al., 2022b), and the incorporation of explicit instructions (Chung et al., 2024; Wei et al., 2022a). However, the quadratic complexity of attention mechanisms within transformer models necessitates significant computational resources to handle long contexts effectively. This has spurred the development of efficient solutions (Dai et al., 2019; Wu et al., 2022; Bulatov et al., 2022) that process long contexts via short context windows, much like how humans process information by reading a book chapter by chapter.

A central question underlying these solutions is: *how can we effectively model and manage memory to bridge the gaps between these short context windows over long context?*

Memory fundamentally revolves around compressing and storing information for future utilization, all within the constraints of limited capacity. The Long Short-Term Memory (LSTM) architecture (Hochreiter & Schmidhuber, 1997) tackles this by recurrently compressing historical information into hidden states after processing each token. With the rise of Transformer models (Vaswani et al., 2017) dominating the language modeling landscape, recent memory designs have shifted towards utilizing Transformers to process a context window, thereby moving the focus of memory management from the token level to the context window level.

Transformer-XL (Dai et al., 2019) employs a caching mechanism to store multi-layer key-value (KV) pairs from the preceding window as memory. Memorizing Transformer (Wu et al., 2022) builds upon this foundation by incorporating a dedicated layer to memorize all KV pairs from that layer across all prior windows. Meanwhile, Block Recurrent Transformer (Hutchins et al., 2022) and Recurrent Memory Transformer (Bulatov et al., 2022) introduce distinct recurrent compression mechanisms, implemented in a middle layer and at the output, respectively.

In this paper, we introduce MELODI (short for "MEmory with LOw DImension"), an efficient memory architecture designed to handle long contexts despite operating on short context windows (e.g., 512 tokens per window). MELODI integrates both short-term and long-term memory through a compression-based approach. The short-term memory, recurrent in nature and possessing low capacity, spans multiple transformer layers, progressively compressing context tokens and prior memory at each layer. In contrast, the long-term memory, incremental and high-capacity, resides within a single transformer layer. It maintains a record of the entire history by further compressing each context window and concatenating them. Both short-term and long-term memory are seamlessly incorporated into a multi-layer transformer model using a "sandwich" structure (see Figure 1), incurring negligible additional parameters.

MELODI demonstrates strong performance on various long-context datasets. For instance, utilizing a 13-layer transformer network with 1024 embedding dimensions and 512-token context windows, MELODI achieves perplexity scores of 10.44 and 2.11 on PG-19 (T5 vocabulary) and arXiv Math (Meena vocabulary), respectively. This represents a clear improvement over the Memorizing Transformer (10.62 on PG-19, 2.14 on arXiv) with dense attention (as opposed to top-k attention), while significantly reducing memory usage by a factor of 8. Furthermore, ablation studies confirm the complementary nature of short-term and long-term memory in MELODI, highlighting their synergistic contribution to an efficient and effective memory architecture.

## 2 RELATED WORK

**Memory in language models:** Long Short-Term Memory (LSTMs) (Hochreiter & Schmidhuber, 1997) use token-level recurrence to compress prior context into a state vector, which is a limited form of memory. With the advent of Transformers (Vaswani et al., 2017), the focus has shifted to memory mechanisms operating at the level of the context window; this shift allows blocks of tokens (i.e. all tokens within the window) to be processed in parallel. The Block Recurrent Transformer (Hutchins et al., 2022) and Recurrent Memory Transformer (Bulatov et al., 2022) integrate recurrent mechanisms inspired by LSTMs into the Transformer architecture, but the recurrence is over blocks, rather than individual tokens. Transformer-XL (Dai et al., 2019) introduces a caching mechanism to store key-value (KV) pairs from the preceding context window as a form of short-term memory. Memorizing Transformer (Wu et al., 2022) uses a large cache to store KV pairs for long-term memory, but only in one layer. MemoryLLM (Wang et al., 2024) incorporates long-term memory in every layer, incurring a substantial memory overhead. Infini-Transformer (Munkhdalai et al., 2024) explore the use of additional memory like Hopfield Networks (Hopfield, 1982; Ramsauer et al., 2021). LONGMEM (Wang et al., 2023) improves over Memorizing Transformer by introducing a SideNet for memory retrieval and fusion. You Only Cache Once (Sun et al., 2024) shows that long-term KV cache is reuable for the latter half of the network, significantly improving pre-filling efficiency by enabling early exit. MELODI, in contrast, integrates integrates both short-term and long-term memory into a transformer model via compression.

**Compression:** Recent work has explored using summary tokens for compression in Transformers (Rae et al., 2019; Bulatov et al., 2022; Chevalier et al., 2023; Ge et al., 2024). Recurrent Memory

Transformer (Bulatov et al., 2022) utilizes the output of summary tokens recurrently as short-term memory. AutoCompressor (Chevalier et al., 2023) aggregates summary tokens across segments to generate a summary representation for long documents used in retrieval tasks. Gisting (Mu et al., 2024) applies this technique to compress long prompts. The In-context Autoencoder (Ge et al., 2024) further incorporates LoRA fine-tuning (Hu et al., 2021) for context compression, while Transformer-FAM (Hwang et al., 2024) introduces feedback attention to enhance performance. Unlike these methods that compress input tokens, MELODI compresses network activations over multiple layers.

**Extending context length:** Recent research has demonstrated promising progress in scaling the context length of language models. To mitigate the cost of attention mechanisms over long contexts, LongLoRA (Chen et al., 2024), PCW (Ratner et al., 2023), and Landmark-Attention (Mohtashami & Jaggi, 2023) employ sparse local attention for efficient fine-tuning. Position-Interpolation (Chen et al., 2023) and YaRN (Peng et al., 2023) extend context size by modifying the RoPE embedding scheme (Su et al., 2023). Xiong et al. (2023) provide a practical recipe for extending LLAMA2 (Touvron et al., 2023) to handle up to 32,768 tokens, while Fu et al. (2024) further explore data-centric approaches for extending context length through lightweight continual pretraining.

# 3 OUR METHOD: MELODI

MELODI focuses on efficiently comprehending *long contexts* while still using *short context windows*, thus circumventing the quadratic complexity associated with attention mechanisms over long sequences. This approach necessitates a memory design that not only ensures smooth transitions between windows but also preserves crucial information from all preceding windows.

## 3.1 ARCHITECTURE OVERVIEW

**Design principle:** The core principle of MELODI is to represent short-term and long-term memory through a hierarchical compression scheme. Specifically, the short-term memory recurrently compresses context across multiple transformer layers (e.g., condensing a 512-token context window into 128 memory tokens). This process not only facilitates seamless transitions between context windows but also aggregates information across them, effectively functioning as a fixed-size multi-layer long short-term memory (LSTM) mechanism (Hochreiter & Schmidhuber, 1997). Furthermore, each context window undergoes additional compression within a middle layer and is then concatenated into a long-term memory. This long-term memory retains essential information from the entire history, thus compensating for any potential forgetting in the short-term memory. Both short-term and long-term memory are seamlessly integrated into a transformer-based language model, enabling the comprehension of long contexts even under the constraint of short context windows.

**Sandwich architecture:** MELODI's network architecture (Figure 1) features a "sandwich" structure, with a long-term compression layer inserted between multiple recurrent short-term compression layers. Both layer types utilize a standard transformer block (attention and feed-forward network) with tailored compression modifications. The short-term layers recurrently compress the current context window and update short-term memory, while the long-term layer further compresses information and appends it to long-term memory.

**Terminology:** In the remainder of this paper, we adopt the following notation. We use $k$ to index context windows and $l$ to index transformer layers. Within the $l^{th}$ layer of the $k^{th}$ context window, the input context tokens are represented by $x_k^{l-1}$. The output context tokens, denoted as $x_k^l$, serve as input for the subsequent layer. The input and output of the short-term memory are $z_{k-1}^l$ and $z_k^l$, respectively. The long-term memory preceding window $k$ is denoted as $m_{1:k-1}$. Note that we omit the subscript $l$ for the long-term memory since it resides within a single long-term layer. Next, we will discuss both short-term and long-term layers in detail.

## 3.2 SHORT-TERM MEMORY: MULTI-LAYER RECURRENT COMPRESSION

The short-term memory is distributed across multiple short-term layers (see Figure 1). This subsection delves into the specifics of the short-term layer, using the $l^{th}$ layer as an illustrative example for processing the $k^{th}$ context window. The short-term layer serves a dual purpose: (a) transforming context tokens $x_k^{l-1}$ via a transformer block (yielding output $x_k^l$), and (b) recurrently compressing

the current context window into the short-term memory $z_k^l$. It accomplishes this by updating both context tokens and short-term memory through a shared transformer block, albeit along separate pathways. As visualized in Figure 2, context tokens traverse vertically across layers (from $x_k^{l-1}$ to $x_k^l$), whereas short-term memory flows horizontally across context windows (from $z_{k-1}^l$ to $z_k^l$). To enable inter-layer communication within the short-term memory, we introduce summary tokens $u_k^l$ that propagate through the layers. We elaborate on the key components below.

**Short-term memory** $z_k^l$**:** The short-term memory (illustrated in Figure 2) is implemented as a sequence of length $S$ of vectors, each having the same dimensionality as context tokens (e.g., 1024 channels). Notably, the number of short-term memory vectors is substantially smaller than the length of the context window (e.g., 128 memory tokens per window of 512 context tokens). Within each context window, the short-term memory serves initially as a previous context for auto-regressive prediction of subsequent context tokens (except for the first window in which the short-term memory is empty). It is then updated by compressing and incorporating information from the context tokens within the current window.

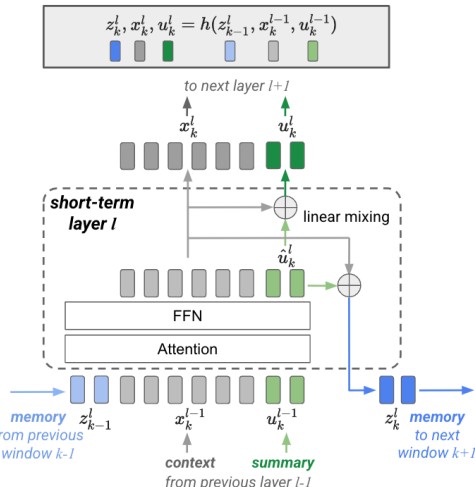

Figure 2: **Short-term layer.** The figure illustrates the processing of the $k^{th}$ context window at the $l^{th}$ short-term layer. It takes the memory from the previous window $z_{k-1}^l$ and the current context/summary $(x_k^{l-1}, u_k^{l-1})$ from the previous layer as input. The short-term layer adds two linear token mixers (Tolstikhin et al., 2021) on top of a standard transformer layer (including attention and FFN) to separate the summary for the next layer $u_k^l$ and the memory for the next window $z_k^l$. Best viewed in color.

**Transforming context tokens:** The context tokens $x_k^l$ are generated through causal attention to both (a) the preceding short-term memory $z_{k-1}^l$ and (b) preceding tokens within the current context window. This attention mechanism is followed by the application of a feed-forward network (FFN). Relative position embeddings are employed for both the context tokens $x_k^{l-1}$ and the preceding short-term memory $z_{k-1}^l$. Mathematically, this can be represented as:

$$x_k^l = \mathcal{T}(x_k^{l-1}|z_{k-1}^l), \tag{1}$$

where $\mathcal{T}(x|z)$ indicates applying a transformer block on $x$ for a given preceding context $z$.

**Recurrent compression:** Beyond transforming context tokens, the short-term layer also recurrently compresses the current context window into short-term memory. Similar to the approach in RMT (Bulatov et al., 2022) and AutoCompressors (Chevalier et al., 2023), this compression is achieved by appending summary tokens $u$ after context tokens $x$ and passing the combined sequence through the transformer block. Consequently, the resulting summary tokens compresses both the preceding short-term memory and the current context window via attentional pooling (Lee et al., 2019), expressed as: $\hat{u}_k^l = \mathcal{T}(u_k^{l-1}|z_{k-1}^l, x_k^{l-1})$, where the input summary tokens $u_k^{l-1}$ originate from the previous layer (refer to Figure 2). We use $\hat{u}_k^l$ (instead of $u_k^l$) to denote an intermediate result that is further processed in the subsequent summary branching step. Both context and summary tokens can be updated simultaneously within a single transformer operation: $x_k^l, \hat{u}_k^l = \mathcal{T}(x_k^{l-1}, u_k^{l-1}|z_{k-1}^l)$. Relative position embeddings are applied on the short-term memory $z_{k-1}^l$, context $x_k^{l-1}$, and summary $u_k^{l-1}$, while a causal mask is applied on the combined sequence of $x_k^{l-1}$ and $u_k^{l-1}$.

Summary tokens (containing $U$ tokens) are initialized from learnable embeddings (prior to the first layer) and set to the same length as the short-term memory ($U = S$). Propagating through all layers, they facilitate inter-layer communication within the short-term memory. Moreover, branching summary tokens both upwards to the next layer and rightwards to the next window improves performance, a strategy we will discuss in more detail subsequently.

**Summary branching:** We employ distinct linear token mixers (Tolstikhin et al., 2021) on the context and summary tokens to generate separate summary tokens for the subsequent layer and short-

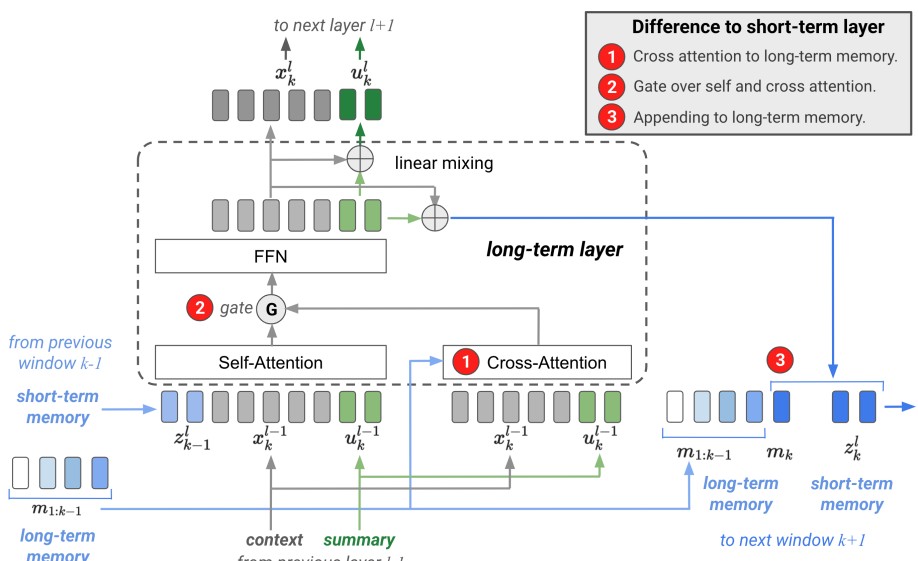

Figure 3: **Long-term layer.** The long-term layer adds three components on top of the short-term layer (see Figure 2). Firstly, it introduces a long-term memory $m_{1:k-1}$ by caching the compressed key-value pairs and allows the current context/summary $(x_k^{l-1}, u_k^{l-1})$ to cross attend to them. Secondly, the self-attention and cross-attention are integrated via gating. Finally, the linear token mixing output additional compressed tokens and appends their key-value pairs $m_k$ into the long-term memory (as $m_{1:k}$) for the next window. Best viewed in color.

term memory tokens for the next window. Unlike channel mixing, a linear token mixer linearly combines the $M_i$ input tokens across each channel to produce $M_o$ output tokens with the same dimensionality by using an $M_i \times M_o$ matrix. The resulting summary and memory tokens exhibit distinct linear combination of context $x_k^l$ and summary tokens $\hat{u}_k^l$, implying divergent compression flows across layers and windows. Since the summary and short-term memory share the same number of tokens ($S$), each mixer comprises $(W + S) \times S$ parameters, where $W$ represent the number of context tokens per window. This parameter count is negligible for short context windows. For instance, with a context window of 512 tokens and 128 summary tokens, the two mixers collectively require $(512+128) \times 128 \times 2 = 164K$ parameters, constituting a mere 1.3% of a transformer block with 1024 dimensions.

**Summary:** The short-term memory layer can be succinctly represented as a function $z_k^l, x_k^l, u_k^l = h(z_{k-1}^l, x_k^{l-1}, u_k^{l-1})$. This function transforms context tokens $x_k^{l-1}$ and summary tokens $u_k^{l-1}$ upward to the next layer (from $l - 1$ to $l$) while simultaneously propagating short-term memory $z_{k-1}^l$ rightward to the next window (from $k - 1$ to $k$). Built upon a standard transformer block, this layer introduces negligible additional parameters through the summary branching mechanism.

**Relation to Block Recurrent Transformer (BRT) (Hutchins et al., 2022):** Like MELODI, BRT also combines a block-wise recurrence mechanism with a Transformer-XL style KV-cache. However, BRT only uses recurrence in a single "memory" layer, while MELODI adds recurrence to all layers. The MELODI recurrence mechanism relies on summary tokens, which do not require any new parameters, while BRT uses a separate "horizontal" transformer block for recurrence instead, which does introduce new parameters. Finally, MELODI builds summaries incrementally over multiple layers (by passing summary tokens), while BRT recurrent state is visible only to a single layer.

### 3.3 LONG-TERM MEMORY: SINGLE-LAYER MEMORIZING COMPRESSED KEY-VALUE PAIRS

While short-term memory facilitates smooth transitions between context windows, the inherent limitation of its capacity can lead to the inevitable loss of information, particularly for contexts located further back in the sequence. In this subsection, we utilize long-term memory to retain information from all previous windows, thereby alleviating the forgetting inherent in short-term memory. The key idea involves further compressing the context window and storing the compressed representations across the entire history.

Table 1: **Long-term vs. short-term memory.** In our notation, $L$ and $S$ represent the number of long-term and short-term tokens per network layer, respectively. Note that long-term has fewer tokens ($L < S$). $Q_{max}$ denotes the maximum number of windows encompassed by the long-term memory queue. $N$ signifies the total number of network layers.

| PROPERTY | LONG | SHORT |
|---|---|---|
| **Number of layers** | *single* | *multiple* |
| **Update per window** | *incremental* | *recurrent* |
| **Capacity** | $L \times Q_{max}$ | $S \times N$ |

Table 2: **Baseline implementations.** Our re-implementations utilize a cosine decay learning rate schedule (replacing inverse square root decay) and dense cross-attention for the Memorizing Transformer (replacing top-k attention). This results in improved performance compared to prior reported results.

| Baseline Method | **PG-19** (T5) | | **arXiv** (Meena) | |
|---|---|---|---|---|
| | prior | **our** | prior | **our** |
| Transformer XL | 11.96 | **11.54** | 2.67 | **2.61** |
| Block Recurrent | 11.55 | **10.98** | 2.36 | **2.26** |
| Memorizing Trans. | 11.62 | **10.74** | 2.31 | **2.15** |

**Further compression:** In contrast to the short-term memory, which compresses information across multiple layers, the long-term memory achieves a higher compression rate within *a single layer*. Specifically, it compresses a context window into $L$ long-term tokens at a designated middle layer (refer to Figure 1), where $L$ is less than the number of short-term tokens per layer ($L < S$). Illustratively, we might compress a context window of 512 tokens into $S = 128$ short-term tokens at each layer, but further condense it into $L = 64$ long-term tokens at a single layer.

**Storing long-term memory:** The key-value (KV) pairs of long-term tokens are sequentially stored in a first-in-first-out (FIFO) queue with a maximum capacity of $Q_{max}$ windows. Consequently, the long-term memory can hold up to $L \times Q_{max}$ KV pairs. For contexts shorter than $Q_{max}$ windows, a compressed representation of the entire prior history is preserved. For longer documents exceeding $Q_{max}$ windows, a substantial portion of the recent history ($Q_{max}$ windows) is still retained. We opt to store KV pairs (rather than the tokens themselves) because they are repeatedly utilized in cross-attention mechanisms for subsequent context windows, a point we will elaborate on later.

**Long-term layer:** Figure 3 illustrates a long-term layer, which builds upon a short-term layer but incorporates three key additions. First, it introduces a long-term memory (denoted as $m_{1:k-1}$ prior to the $k^{th}$ context window) and enables the current context tokens $x_k$ and summary tokens $u_k$ to cross-attend to it. Second, the cross-attention shares parameters with the self-attention, and their results (cross attention: $\mathcal{A}_x$, self-attention: $\mathcal{A}_s$) are combined through a gating mechanism using a learnable scalar $\alpha$ per attention head, formulated as $\alpha \mathcal{A}_x + (1 - \alpha)\mathcal{A}_s$. Finally, an additional linear token mixer is introduced to generate long-term tokens for the current window, and their key-value (KV) pairs $m_k$ are appended to the long-term memory. This token mixer comprises $(W + U) \times L$ parameters, where $W$, $U$ and $L$ represent the number of context, summary and long-term memory tokens, respectively. It is notably smaller than the mixers in the short-term layer because: (a) $L$ is less than $S$ (the number of short-term memory tokens), and (b) it is present in only one layer.

**Long-term vs short-term memory:** Table 1 provides a comparative overview of the long-term and short-term memory mechanisms. While short-term memory operates recurrently across $N$ layers with $S$ tokens per layer, long-term memory functions incrementally in one layer, spanning $Q_{max}$ windows with $L$ tokens per window. To illustrate, consider a 13-layer transformer model processing context windows of 512 tokens each, with embedding dimension 1024. If we employ $S = 128$ short-term tokens per layer and $L = 64$ long-term tokens per window, with a maximum capacity of $Q_{max} = 128$ windows for the long-term memory, the short-term and long-term memory caches would have capacities of 1.7M ($128^{tokens} \times 1024^{dim} \times 13^{layers}$) and 16.8M ($64^{pairs} \times 2^{tokens/pair} \times 1024^{dim} \times 128^{windows}$) floats, respectively. It's worth noting that, for computational efficiency, we store key-value (KV) pairs in the long-term memory, whereas tokens are stored directly in the short-term memory.

**Relation to Memorizing Transformer (MT) (Wu et al., 2022):** Similar to Memorizing Transformer, MELODI incorporates key-value pairs from a middle layer into its long-term memory. Experiments corroborate the finding in MT that employing additional long-term layers yields only incremental gains. However, a key distinction lies in the fact that MELODI stores *compressed* KV pairs, rather than the KV pairs of context tokens directly, as in MT. This modification substantially reduces the size of the long-term memory. For instance, when compressing a context window of 512 tokens into 64 long-term tokens, MELODI achieves an *8-fold reduction* in long-term memory size.

Table 3: **Comparisons with baselines on three datasets**. The table reports average token-level perplexities for various models on three datasets. All models were trained under the same settings (e.g. segment length 4096, context window 512, 500k training steps). Three MELODI configurations were used: $S_{192} + L_{32}$, $S_{128} + L_{64}$, and $S_{192} + L_{96}$. For instance, $S_{192} + L_{32}$ indicates $S = 192$ short-term and $L = 32$ long-term tokens per context window. All MELODI models utilized a long-term memory spanning 128 context windows. MELODI $S_{192} + L_{32}$ outperformed Transformer XL and Block Recurrent Transformer while consuming less memory. Notably, MELODI $S_{192} + L_{96}$ clearly surpassed Memorizing Transformer, using only a fifth of its memory.

| MODEL | MEMORY | | | PG19 | | | arXiv | | C4(4K+) |
|---|---|---|---|---|---|---|---|---|---|
| | All | Short | Long | Meena | T5 | Custom | Meena | Custom | Custom |
| Transformer XL | 13.6M | 13.6M | 0M | 8.65 | 11.41 | 12.42 | 2.60 | 3.22 | 18.22 |
| Block Recurrent | 13.1M | 13.1M | 0M | 8.30 | 10.98 | 11.90 | 2.26 | 2.70 | 17.82 |
| **MELODI** $S_{192}+L_{32}$ | **11.0M** | 2.6M | 8.4M | **8.08** | **10.51** | **11.47** | **2.12** | **2.54** | **17.55** |
| Memorizing Trans. | 147.8M | 13.6M | 134.2M | 8.07 | 10.62 | 11.53 | 2.14 | 2.56 | 17.37 |
| **MELODI** $S_{128}+L_{64}$ | **18.5M** | 1.7M | 16.8M | 8.06 | 10.44 | 11.42 | 2.11 | 2.52 | 17.53 |
| **MELODI** $S_{192}+L_{96}$ | 27.8M | 2.6M | 25.2M | **7.91** | **10.29** | **11.27** | **2.09** | **2.49** | **17.25** |

## 4 EXPERIMENTAL RESULTS

We evaluate MELODI on three long-context datasets: PG19 (Rae et al., 2019), arXiv Math (Wu et al., 2022), and C4 (Raffel et al., 2020a), using the standard auto-regressive language modeling task, where the objective is to predict the next token in a sequence. All models are trained from scratch, and we report the average perplexity on the respective test sets as our evaluation metric.

### 4.1 DATASETS

**PG19:** The PG19 dataset (Rae et al., 2019) consists of 28,752 English books published before 1919, averaging around 68,972 tokens per book. We utilize three 32k vocabularies: (a) Meena (Thoppilan et al., 2022), (b) T5 (Raffel et al., 2020b) and (c) a custom SentencePiece vocabulary (Kudo & Richardson, 2018) trained specifically on PG19.

**arXiv Math:** The arXiv dataset (Wu et al., 2022) comprises technical math papers from arXiv, with token counts comparable to PG19 due to LaTeX's use of special characters, resulting in smaller subwords. We use a 32k Meena vocabulary (Thoppilan et al., 2022) and a 32k custom vocabulary.

**C4(4K+):** The C4 dataset (Raffel et al., 2020a) is a large collection of internet documents. To emphasize long documents where memory is crucial, we filter out those with fewer than 4,096 tokens and utilize a 32k custom vocabulary.

### 4.2 SETUP

We utilized a decoder-only transformer architecture (with 12 or 13 layers), incorporating both short-term and long-term memory caches. The model had an embedding size of 1024, 8 attention heads with a dimensionality of 128 each, and an FFN hidden layer of size 4096. All models were implemented in JAX and Flax and trained from scratch for 500k steps on 32 TPU cores. Further training details are provided in the Appendix C.3.

During training, each long document was segmented into 4096-token chunks to facilitate batch processing. These chunks were then organized into training batches, each comprising 8 context windows of 512 tokens. In our ablation study, MELODI's default configuration compressed each context window into $S$=128 short-term memory tokens per layer and $L$=64 long-term memory tokens.

### 4.3 COMPARISON WITH BASELINES

**Baselines:** We benchmark MELODI against three well-established prior works: Transformer XL (Dai et al., 2019), Block Recurrent Transformer (Hutchins et al., 2022), and Memorizing Transformer (Wu et al., 2022). To ensure a fair comparison, we re-implement these baselines within our framework and evaluate them under identical settings. Our re-implementations of the baseline mod-

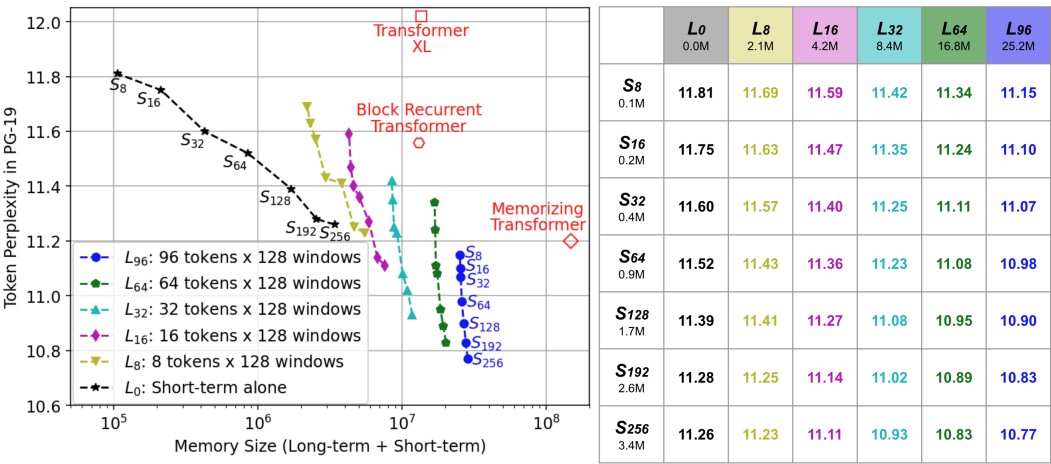

Figure 4: **Ablation of memory size on PG-19.** The token perplexity is reported for various combinations of short-term and long-term memory sizes. Each curve represents a fixed size of long-term memory, with points along the curve indicating different short-term memory sizes. For example, the blue curve ($L_{96}$), uses 96 long-term tokens per window over 128 windows, totaling 12,288 tokens. Each point on this curve represents a different short-term memory size (e.g. $S_8$ denotes 8 short-term tokens per context window). Memory size is measured by the number of floating-point numbers (floats). For instance, $L_{96}$ stores 12,288 long-term key-value (KV) pairs, each with 1024 dimensions, resulting in a total of $12,288 \times 1024 \times 2 = 25.2$ million floats. The table on the right provides the perplexity results for each point on the left plot, using matching colors. These results highlight that long-term and short-term memories play complementary roles, and increasing either type's capacity improves performance. Notably, MELODI achieves superior performance compared to baselines like Transformer XL, Block Recurrent Transformer and Memorizing Transformer while utilizing fewer memory resources. Best viewed in color.

els achieve superior performance compared to the results reported in their original papers (see Table 2). This improvement can be primarily attributed to two key factors: (a) using cosine decay learning rate schedule (Hoffmann et al., 2022) instead of the inverse square root decay, (b) using dense cross-attention instead of top-k attention for the Memorizing Transformer. Our re-implementations provide stronger baselines against which to evaluate MELODI's effectiveness.

**Comparisons:** Table 3 presents a comparison of MELODI against three baseline models (Transformer XL, Block Recurrent Transformer, and Memorizing Transformer) across three datasets. We evaluate three MELODI configurations: $S_{192} + L_{32}$, $S_{128} + L_{64}$, and $S_{192} + L_{96}$, where $S$ and $L$ denote the number of short-term and long-term tokens per context window, respectively. All models (MELODI and baselines) utilize a 13-layer transformer architecture, except for Block Recurrent Transformer, which inserts a block recurrent layer into a 12-layer transformer, ensuring a similar parameter count for all models. Results demonstrate both the impact on LLM performance and efficiency of MELODI's memory mechanisms.

*Impact on LLM performance:* Compared to Transformer XL, a transformer-based language model with a KV cache of the previous context window, MELODI $S_{192}+L_{32}$ uses about the same amount of memory but significantly outperforms Transformer XL in terms of perplexity. For instance, on the PG-19 dataset with the T5 vocabulary, MELODI $S_{192}+L_{32}$ achieves a perplexity of 10.51, compared to 11.41 for Transformer XL and 10.98 for Block Recurrent Transformer (enhanced Transformer XL with recurrent memory), demonstrating a clear improvement. This highlights the effectiveness of MELODI's hierarchical memory mechanism in capturing and utilizing long-range context.

*Efficiency:* Compared to Memorizing Transformer, MELODI variants ($S_{128} + L_{64}$ and $S_{192} + L_{96}$) consistently achieves comparable or slightly better perplexity across all datasets, but uses approximately 8 and 5 times less memory respectively.

These trends remain consistent across different network depths (12-layer and 13-layer), as shown in Table 5 in Appendix D.1. These results collectively highlight MELODI's efficacy and efficiency as a memory architecture for language models.

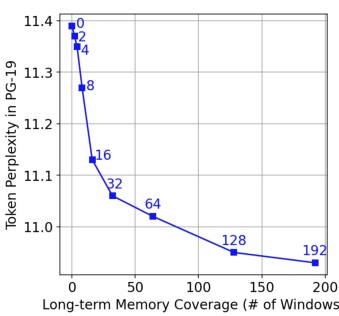

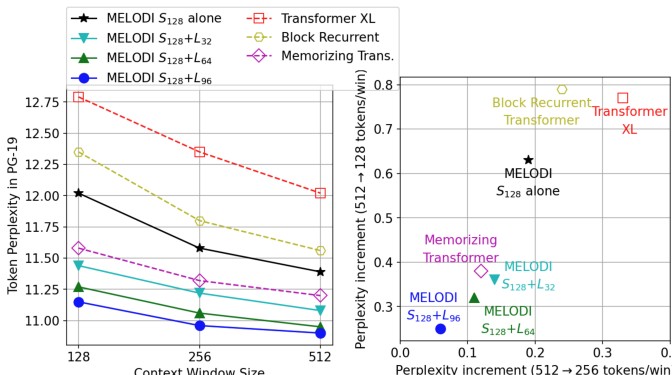

Figure 5: **Long-term memory coverage.** The coverage metric indicates the number of preceding context windows (512 tokens per window) spanned by the long-term memory. All data points use $L$=64 long-term tokens per window, but vary in long-term memory capacity. For instance, '192' denotes storing information from previous 192 context windows, with effective context length $192 \times 512$=96k tokens. Perplexity improves as the long-term memory coverage increases, especially from 4 to 32 previous windows.

Figure 6: **Shorter context windows.** (*Left*): *Perplexity with varying context window sizes (512, 256, 128 tokens).* The number of short-term and long-term memory tokens in MELODI is proportionally adjusted ($\times \frac{1}{2}$, $\times \frac{1}{4}$, respectively) to ensure consistent long-term memory coverage. Even with smaller windows, MELODI with appropriate memory allocation (e.g. $S_{128}+L_{64}$) consistently outperforms baselines. (*Right*): *Perplexity increase due to window size reduction.* The $x$ and $y$ axes represent the perplexity increment when reducing the window size from 512 to 256 and 128 tokens, respectively. Models with long-term memory (i.e. MELODI variants with long-term memory and Memorizing Transformer) exhibit significantly less performance degradation (smaller perplexity increments) compared to those relying solely on short-term memory (MELODI with only short-term memory, Transformer XL, and Block Recurrent Transformer).

## 4.4 ABLATIONS

We conduct an ablation study of MELODI using the default configuration ($S_{128}+L_{64}$) with 128 short-term and 64 long-term tokens per context window. The long-term memory spans 128 context windows, and the transformer architecture consists of 13 layers. All models are trained on the PG-19 dataset with the T5 vocabulary for 200k steps.

**Complementary roles of short-term and long-term memory:** Figure 4 illustrates how the sizes of both short-term and long-term memory jointly influence perplexity. Each curve represents a fixed long-term memory size, with varying short-term memory sizes depicted by points along the curve. With the exception of the black curve, which solely utilizes short-term memory, all other curves incorporate long-term memory spanning 128 context windows. The figure demonstrates that increasing either short-term or long-term memory capacity leads to improved perplexity, highlighting their complementary roles in performance. Notably, by judiciously selecting memory sizes (e.g., $S_{192}$ for short-term and $L_{32}$ for long-term), we can outperform the Memorizing Transformer while utilizing less memory than Transformer XL and Block Recurrent Transformer.

**Impact of long-term memory coverage:** In contrast to the previous ablation, we now maintain a fixed number of short-term ($S = 128$) and long-term tokens ($L = 64$) per context window while varying the number of windows encompassed by the long-term memory. Figure 5 demonstrates that performance improves as the long-term memory covers a wider range of context windows. Interestingly, incorporating long-term memory for only the preceding 2 or 4 windows yields marginal perplexity improvements, suggesting that recent context is already effectively captured by the short-term memory. However, performance gains accelerate as the long-term memory expands to encompass up to 32 windows, after which the improvements level off. This observation indicates that while the middle and distant history are beneficial for language modeling, they are not adequately retained in the short-term memory. These findings further underscore the complementary nature of short-term and long-term memory mechanisms.

**Performance with shorter context windows:** Figure 6 examines the impact of reducing context window size on model performance. The plot on the left displays perplexity scores for context window sizes of 512, 256, and 128 tokens. The number of short-term and long-term memory to-

Table 4: **Summary branching.** Summary branching provides a consistent gain of approximately 0.3 in perplexity, both with (column ST+LT) and without (column ST) long-term memory.

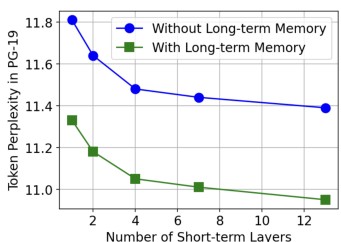

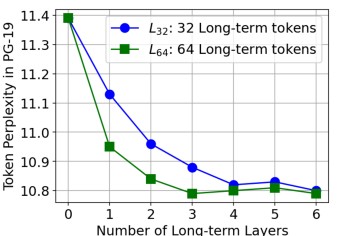

| Branching | ST | ST+LT |
|---|---|---|
| No | 11.68 | 11.24 |
| Yes | **11.39** | **10.95** |

Figure 7: **Number of short-term layers.** Perplexity improves as the number of short-term layers increases.

Figure 8: **Number of long-term layers.** A single layer with sufficient long-term tokens is adequate.

kens in MELODI is proportionally adjusted ($\times\frac{1}{2}$, $\times\frac{1}{4}$, respectively) to ensure consistent long-term memory coverage across different window sizes. Even with shorter context windows, MELODI with appropriate memory allocation (e.g., $S_{128} + L_{64}$) consistently outperforms the baseline models.

The plot on the right illustrates the increase in perplexity resulting from reducing the context window size. Notably, models incorporating long-term memory (MELODI variants with long-term memory and Memorizing Transformer) exhibit significantly less performance degradation (smaller perplexity increments) compared to models relying solely on short-term memory (MELODI variant with only short-term memory, Transformer XL, and Block Recurrent Transformer). This highlights the greater robustness of long-term memory mechanisms to reductions in context window size.

**Number of short-term layers:** We investigate the effect of varying the number of short-term layers on model performance. For a given layer count (e.g., 4 layers), the short-term layers are uniformly distributed throughout the network (e.g., layers 1, 5, 9, and 13). To disable short-term memory within a layer, we remove (a) the attention mechanism to the preceding short-term memory and (b) the linear token mixer responsible for updating the short-term memory. Figure 7 shows that perplexity improves rapidly as the number of short-term layers increases from 1 to 4, after which the gains diminish. This observation supports our utilization of multiple layers for effective short-term memory modeling. However, it also suggests that disabling short-term memory in half of the layers offers a more efficient approach with negligible performance degradation.

**Number of long-term layers:** Figure 8 shows the impact of the number of long-term layers on perplexity for two configurations: $L_{64}$ and $L_{32}$, which compress each window into 64 and 32 tokens, respectively. Fewer tokens ($L_{32}$) require more long-term layers for stable perplexity: $L_{32}$ needs 4 layers to reach a plateau, while $L_{64}$ only needs 3. Notably, with additional tokens ($L_{64}$), the first long-term layer contributes most significantly to performance improvement. It provides 73% of the total gain achieved with 6 layers. This suggests that a single long-term layer with sufficient tokens offers a good balance between performance and efficiency, as adding more layers increase memory and computational costs with diminishing returns.

**Summary branching:** Table 4 examines the effect of summary branching on perplexity, both with (ST+LT) and without (ST) long-term memory. Summary branching consistently yields a perplexity improvement of approximately 0.3, indicating distinct summary information flow across network layers and context windows.

## 5 CONCLUSION

In this work, we have introduced MELODI, a novel memory architecture designed to address the challenges of long document processing within the constraints of short context windows. The core innovation of MELODI lies in its hierarchical compression approach, wherein short-term memory facilitates smooth transitions between context windows through recurrent compression across multiple layers, and long-term memory preserves crucial information from the entire history by performing further compression and aggregation within a single middle layer. Our empirical evaluations on multiple long-context datasets have validated MELODI as an efficient and effective solution. The success of MELODI underscores the potential of hierarchical memory compression for tackling the complexities of long document processing. We anticipate that further research in this direction will enhance long context understanding and generation over multiple modalities.

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

# A   LIMITATIONS AND FUTURE WORK

## A.1   LIMITATIONS

A primary limitation of our current study is the focus on training models from scratch. We acknowledge the importance of evaluating MELODI on downstream tasks and with pre-trained language models, which often have fixed context window sizes. However, adapting MELODI to such settings requires larger models and more extensive pre-training.

Furthermore, our evaluation primarily relies on perplexity as a performance metric. While perplexity provides a useful measure of language modeling capability, it may not fully capture real-world performance (Hu et al., 2024; Fang et al., 2024), the gap is somewhat emphasized for smaller models (Muhlgay et al., 2024).

While MELODI does not introduce additional parameters, the summary tokens within its short-term and long-term layers, along with the cross-attention mechanism in the long-term memory, do increase FLOPs (see Appendix E.3). Consequently, training time is extended when compared to Transformer XL (Dai et al., 2019). However, it only slightly increases test time (see Appendix E.4).

## A.2   FUTURE WORK

In future work, we plan to address these limitations by:

**Integrating MELODI with pre-trained models:** We will explore techniques like LoRA (Low-Rank Adaptation) to efficiently integrate MELODI's short-term and long-term memory mechanisms into pre-trained language models. This will enable us to leverage the benefits of pre-training and evaluate MELODI's effectiveness on downstream tasks like summarization and question answering.

**Evaluating on downstream tasks:** By adapting MELODI to pre-trained models, we can conduct a comprehensive evaluation of its effectiveness on a broader range of long-context benchmarks and downstream tasks. This will facilitate direct comparisons with state-of-the-art baselines and provide a more complete picture of MELODI's capabilities.

By addressing these limitations, we aim to provide a more comprehensive evaluation of MELODI and its potential for advancing long-context language modeling.

# B   CONNECTIONS TO RECENT WORK ON LONG-TERM AND SHORT-TERM MEMORY

This section compares MELODI with two recent works that explore different strategies for long-term and short-term memory in language models: LONGMEM (Wang et al., 2023) and YOCO (Sun et al., 2024).

**Comparison with LONGMEM (Wang et al., 2023):** Both LONGMEM and MELODI improve upon the Memorizing Transformer (Wu et al., 2022) but with distinct focuses and methodologies:

- LONGMEM: LONGMEM focuses on improving *prediction accuracy* in a *fine-tuning* setting by introducing a *SideNet* for memory retrieval and fusion. It is compared against the Memorizing Transformer variant that uses *top-k KV pairs* from memory.
- MELODI: MELODI prioritizes *efficiency* in a *train-from-scratch* setting by compressing both long-term and short-term memory. It introduces a negligible number of extra parameters and is compared against the Memorizing Transformer variant that uses *all KV pairs* in memory.

A direct experimental comparison with LONGMEM in the fine-tuning setting is beyond the scope of this paper, as our primary focus is on evaluating MELODI's performance when training from scratch. However, we plan to explore extending MELODI to the fine-tuning of LLMs in future work, which will allow for a comprehensive comparison with LONGMEM and other relevant baselines.

**Comparison with YOCO (Sun et al., 2024):** Both YOCO and MELODI utilize a long-short memory strategy, with short-term memory components spanning multiple layers and long-term informa-

tion cached in a middle layer. However, their core strategies differ, as detailed below. Furthermore, they exhibit distinct properties (reusability vs. compressibility) in their long-term key-value (KV) memory:

- YOCO: YOCO employs sliding-window attention across multiple self-decoder layers for short-term memory and a KV cache in the middle layer for long-term memory. Importantly, YOCO demonstrates that this long-term KV cache is reusable for the latter half of the network, significantly improving pre-filling efficiency by enabling early exit.

- MELODI: MELODI implements a different long-short strategy based on compression. Short-term memory is achieved through recurrent compression of context windows across multiple layers, while long-term memory leverages further compression within a single middle layer. MELODI demonstrates that long-term KV memory is compressible, significantly reducing its size.

YOCO and MELODI enrich the design space for long-short memory models and offer complementary approaches. Combining their strengths could lead to even more effective models. For instance, exploring whether long-term memory can be both reusable (as in YOCO) and compressible (as in MELODI) is a promising avenue for future research. This could enable efficient caching with a compact KV memory footprint.

## C  IMPLEMENTATION DETAILS

### C.1  EFFICIENT PARALLEL TRAINING

To enable efficient parallel training despite recurrent dependencies in short-term and long-term memory, MELODI employs a combined layer-wise and chunk-wise processing strategy. This approach allows for parallel computation within each layer while maintaining the sequential processing required for recurrent memory updates.

**Layer-wise processing and window unrolling:** MELODI utilizes a standard Transformer architecture with multiple layers processed sequentially. Both short-term and long-term memory operations are confined within their respective layers. Within each layer, a training batch covers 8 context windows (512 tokens per window), totaling 4096 tokens. This allows for backpropagation through time (BPTT) across multiple windows. While the windows are processed sequentially due to window-level recurrence, self-attention within each window can be parallelized, ensuring good device utilization with sufficiently large window sizes.

**Parallel and sequential operations:** Within each layer, the computation of query, key, and value matrices for attention is parallelized across all chunks, as these computations are independent of recurrent memory updates. Each layer thus involves two steps: (a) parallel computation of query/key/value matrices, and (b) sequential computation of self-attention, feed-forward networks, and token mixing for memory updates.

This approach contrasts with a chunk-wise then layer-wise strategy, where each chunk must be fully processed by the entire network before the next chunk can begin processing.

### C.2  WINDOW-WISE PROCESSING IN GENERATION

MELODI employs a two-step window-wise (or chunk-wise) processing strategy during generation:

- *Token-wise generation*: Within each context window, MELODI generates tokens sequentially, similar to GPT-like LLMs, utilizing both short-term and long-term memory. This ensures that token generation within a window (or chunk) remains unaffected by window boundaries, preserving the key mechanisms of autoregressive language models.

- *Summary generation*: Upon completing a window (or chunk), MELODI generates all summary tokens in parallel and updates both short-term and long-term memory accordingly.

This approach effectively balances the sequential nature of token generation with the parallel processing of summary information, optimizing generation efficiency.

Table 5: **Comparisons with baselines on three datasets**. The table reports average token-level perplexities for various models on three datasets. All models were trained under the same settings (e.g. segment length 4096, context window 512, 500k training steps). Three MELODI configurations were used: $S_{192} + L_{32}$, $S_{128} + L_{64}$, and $S_{192} + L_{96}$. For instance, $S_{192} + L_{32}$ indicates $S = 192$ short-term and $L = 32$ long-term tokens per context window. All MELODI models utilized a long-term memory spanning 128 context windows. MELODI $S_{192} + L_{32}$ outperformed Transformer XL and Block Recurrent Transformer while consuming less memory. Notably, MELODI $S_{192} + L_{96}$ clearly surpassed Memorizing Transformer, using only a fifth of its memory.

| MODEL | MEMORY | | | PG19 | | | arXiv | | C4(4K+) |
|---|---|---|---|---|---|---|---|---|---|
| | All | Short | Long | Meena | T5 | Custom | Meena | Custom | Custom |
| **12 LAYERS** | | | | | | | | | |
| Transformer XL | 12.6M | 12.6M | 0M | 8.76 | 11.54 | 12.63 | 2.61 | 3.23 | 18.61 |
| Block Recurrent | 12.1M | 12.1M | 0M | 8.47 | 11.12 | 12.11 | 2.27 | 2.73 | 18.27 |
| MELODI $S_{192}+L_{32}$ | **10.8M** | 2.4M | 8.4M | **8.22** | **10.66** | **11.66** | **2.13** | **2.55** | **18.03** |
| Memorizing Trans. | 146.8M | 12.6M | 134.2M | 8.15 | 10.74 | 11.68 | 2.15 | 2.57 | 17.88 |
| MELODI $S_{128}+L_{64}$ | **18.4M** | 1.6M | 16.8M | 8.16 | 10.61 | 11.66 | 2.13 | 2.55 | 18.01 |
| MELODI $S_{192}+L_{96}$ | 27.6M | 2.4M | 25.2M | **8.08** | **10.48** | **11.47** | **2.11** | **2.51** | **17.75** |
| **13 LAYERS** | | | | | | | | | |
| Transformer XL | 13.6M | 13.6M | 0M | 8.65 | 11.41 | 12.42 | 2.60 | 3.22 | 18.22 |
| Block Recurrent | 13.1M | 13.1M | 0M | 8.30 | 10.98 | 11.90 | 2.26 | 2.70 | 17.82 |
| MELODI $S_{192}+L_{32}$ | **11.0M** | 2.6M | 8.4M | **8.08** | **10.51** | **11.47** | **2.12** | **2.54** | **17.55** |
| Memorizing Trans. | 147.8M | 13.6M | 134.2M | 8.07 | 10.62 | 11.53 | 2.14 | 2.56 | 17.37 |
| MELODI $S_{128}+L_{64}$ | **18.5M** | 1.7M | 16.8M | 8.06 | 10.44 | 11.42 | 2.11 | 2.52 | 17.53 |
| MELODI $S_{192}+L_{96}$ | 27.8M | 2.6M | 25.2M | **7.91** | **10.29** | **11.27** | **2.09** | **2.49** | **17.25** |

## C.3 TRAINING SETUP

We use Adafactor optimizer (Shazeer & Stern, 2018) with a learning rate schedule that employs a linear warmup for the first 1000 steps, followed by cosine decay. The maximum and minimum learning rates are set to 0.01 and 0.001, respectively, as recommended in Hoffmann et al. (2022). A dropout rate of 0.05 is applied. All models are trained for 500k steps (200k for ablations) on 32 TPU cores with a batch size of 32 (1 per core).

# D ADDITIONAL EXPERIMENTAL RESULTS

## D.1 MORE COMPARISON WITH BASELINES OVER 3 DATASETS

Table 5 presents a comparison of MELODI against three baseline models (Transformer XL, Block Recurrent Transformer, and Memorizing Transformer) across three datasets, using a consistent segment length of 4096 tokens and a context window size of 512. The evaluation includes both 12-layer and 13-layer transformer architectures to assess the impact of model depth on performance.

Notably, even with fewer layers, MELODI $S_{192}+L_{32}$ consistently outperforms both Transformer XL and Block Recurrent Transformer across all datasets while consuming less memory. For instance, the 12-layer variant of MELODI $S_{192}+L_{32}$ achieves a perplexity of 10.66 on PG-19 (T5 vocabulary), surpassing the 13-layer variants of Transformer XL (11.41) and Block Recurrent Transformer (10.98).

In comparison to the Memorizing Transformer, MELODI $S_{128} + L_{64}$ exhibits slightly improved performance while dramatically reducing memory consumption by a factor of 8. Furthermore, MELODI $S_{192} + L_{96}$ achieves even better perplexity scores across all datasets with a substantial reduction in memory usage exceeding a factor of 5. The improvement is consistent for using both 12-layer and 13-layer transformer architectures. These results collectively highlight MELODI's efficacy and efficiency as a memory architecture for language models.

Table 6: **Comparison of models with larger sizes** evaluated on the PG-19 dataset using the T5 vocabulary. Models are evaluated based on perplexity (lower is better) and memory size. The batch sequence length is 4096, comprising 8 context windows of 512 tokens.

| MODEL | #Layers | Embedding Dim | Model Size | Memory Size | Perplexity $\downarrow$ |
|---|---|---|---|---|---|
| Transformer XL | 12 | 1024 | 151M | 12.6M | 11.54 |
| Memorizing Transformer | 12 | 1024 | 151M | 146.8M | 10.74 |
| **MELODI** $S_{128} + L_{64}$ | 12 | 1024 | 153M | 18.4M | 10.61 |
| **MELODI** $S_{192} + L_{96}$ | 12 | 1024 | 153M | 27.6M | **10.48** |
| Transformer XL | 24 | 1024 | 302M | 25.2M | 10.15 |
| Memorizing Transformer | 24 | 1024 | 302M | 159.4M | 9.45 |
| **MELODI** $S_{128} + L_{64}$ | 24 | 1024 | 306M | 20.0M | 9.30 |
| **MELODI** $S_{192} + L_{96}$ | 24 | 1024 | 309M | 30.0M | **9.16** |
| Transformer XL | 36 | 1024 | 453M | 37.8M | 9.61 |
| Memorizing Transformer | 36 | 1024 | 453M | 172.0M | 8.92 |
| **MELODI** $S_{128} + L_{64}$ | 36 | 1024 | 459M | 21.6M | 8.81 |
| **MELODI** $S_{192} + L_{96}$ | 36 | 1024 | 463M | 32.4M | **8.70** |
| Transformer XL | 16 | 1536 | 453M | 25.2M | 9.69 |
| Memorizing Transformer | 16 | 1536 | 453M | 226.5M | 9.01 |
| **MELODI** $S_{128} + L_{64}$ | 16 | 1536 | 456M | 28.3M | 8.89 |
| **MELODI** $S_{192} + L_{96}$ | 16 | 1536 | 459M | 42.5M | **8.79** |

## D.2 SCALING UP MODEL SIZE

Table 6 presents the results of scaling up MELODI and comparing its performance with baselines on the PG-19 dataset (using the T5 vocabulary). We explored two scaling approaches:

- *Increased depth*: We trained models with 24 and 36 layers, effectively scaling the number of parameters by 2 and 3 times, respectively, compared to the 12-layer base model.

- *Increased width and depth*: We trained a model with 16 layers and an embedding dimension of 1536, which also scales the parameter count by 3 times.

The table shows that MELODI's advantages hold even with larger models, achieving consistently lower perplexity than the Memorizing Transformer while using significantly less memory (approximately 8 and 5 times less). This demonstrates MELODI's effectiveness in scaling to larger architectures while maintaining memory efficiency.

## D.3 SCALING UP CONTEXT LENGTH

To investigate the performance of MELODI with larger context windows, we conducted experiments with batch sequence lengths of 4096 and 8192 tokens and window lengths of 512, 1024, and 2048 tokens. Increasing the context window allows the model to capture longer-range dependencies in the input sequence, which can be crucial for understanding complex language structures and tasks. All models were trained on PG-19 (T5 vocabulary) with 12-layer architectures.

Table D.2 shows that MELODI continues to outperform the Memorizing Transformer with significantly less memory (approximately 8 and 5 times less), even with larger context windows.

Due to limited computational resources, we were unable to train MELODI with context windows exceeding 2048 tokens. Scaling MELODI to even larger context windows presents two primary challenges:

- *Memory capacity*: The memory requirements of MELODI grow with both the context window size and the batch sequence length. With our current hardware and codebase, we are unable to train MELODI with context windows exceeding 2048 tokens. However, this can

Table 7: **Impact of sequence and window length**. This table compares the perplexity and memory usage of Transformer XL, Memorizing Transformer, and MELODI with varying sequence and window lengths on the PG-19 dataset using the T5 vocabulary. All models have 12 layers.

| MODEL | Sequence Length | Window Length | Memory Size | Perplexity ↓ |
|---|---|---|---|---|
| Transformer XL | 4096 | 512 | 12.6M | 11.54 |
| Memorizing Transformer | 4096 | 512 | 146.8M | 10.74 |
| **MELODI** $S_{128} + L_{64}$ | 4096 | 512 | 18.4M | 10.61 |
| **MELODI** $S_{192} + L_{96}$ | 4096 | 512 | 27.6M | **10.48** |
| Transformer XL | 4096 | 1024 | 25.2M | 11.26 |
| Memorizing Transformer | 4096 | 1024 | 159.4M | 10.64 |
| **MELODI** $S_{256} + L_{128}$ | 4096 | 1024 | 20.0M | 10.47 |
| **MELODI** $S_{384} + L_{192}$ | 4096 | 1024 | 30.0M | **10.36** |
| Transformer XL | 8192 | 1024 | 25.2M | 11.18 |
| Memorizing Transformer | 8192 | 1024 | 159.4M | 10.42 |
| **MELODI** $S_{256} + L_{128}$ | 8192 | 1024 | 20.0M | 10.27 |
| **MELODI** $S_{384} + L_{192}$ | 8192 | 1024 | 30.0M | **10.19** |
| Transformer XL | 8192 | 2048 | 50.3M | 10.94 |
| Memorizing Transformer | 8192 | 2048 | 184.5M | 10.38 |
| **MELODI** $S_{512} + L_{256}$ | 8192 | 2048 | 23.1M | 10.20 |
| **MELODI** $S_{768} + L_{384}$ | 8192 | 2048 | 34.6M | **10.10** |

Table 8: **Comparison of MELODI with Transformer XL using varying context window lengths.** The perplexity on the PG-19 dataset using the T5 vocabulary is reported. The batch sequence length is 8192 for all models. Transformer XL is evaluated with context window lengths of 512, 1024, 2048, and 4096, while MELODI uses a context window length of 512.

| MODEL | Sequence Length | Context Window Length | Perplexity ↓ |
|---|---|---|---|
| Transformer XL | 8192 | 512 | 11.40 |
| Transformer XL | 8192 | 1024 | 11.18 |
| Transformer XL | 8192 | 2048 | 10.94 |
| Transformer XL | 8192 | 4096 | 10.65 |
| **MELODI** $S_{128} + L_{64}$ | 8192 | 512 | 10.34 |
| **MELODI** $S_{192} + L_{96}$ | 8192 | 512 | **10.22** |

be addressed by using TPUs with more memory and improving model parallelism, which we plan to investigate in future work.

- *Backpropagation through time (BPTT)*: MELODI's recurrent connections necessitate BPTT, which involves unrolling the network over multiple time steps and further increases memory consumption. Each batch spans 8 context windows by default, allowing for BPTT across multiple windows. While crucial for the recurrent short-term memory, BPTT contributes significantly to memory usage. As shown in Table D.2, using longer batch sequences (8K tokens) with 1K token windows yields better performance than shorter sequences (4K tokens) due to the increased BPTT depth.

## D.4 COMPARISON WITH TRANSFORMER MODELS WITH EXTENDED CONTEXT

To further explore the impact of context window length, we compare MELODI (with a context window of 512 tokens) to a conventional Transformer XL architecture with extended context windows ranging from 512 to 4096 tokens. This comparison allows us to assess the efficiency of MELODI's long-term memory mechanism against a baseline model that relies solely on increasing the context

window within each layer. All models are trained on the PG-19 dataset with the T5 vocabulary for 500k steps, with each batch including 8K tokens to allow backpropagation through time (BPTT) across multiple windows.

Table 8 shows that Transformer XL's performance improves with increasing context length, as expected. Notably, MELODI (512 tokens) outperforms Transformer XL even with an 8 times longer context window (4096 tokens), demonstrating the effectiveness of MELODI's approach in capturing long-range dependencies.

### D.5 NEW TASK: MASKED NEXT TOKEN PREDICTION

We introduce a novel task designed to further evaluate MELODI's capacity for handling long-range dependencies. Similar to our primary evaluation, we use next token prediction and measure performance with perplexity. However, we introduce a key challenge: ***masking out a portion of the input tokens***. Unlike BERT's (Devlin et al.) bidirectional masking approach, this new task employs a unidirectional, left-to-right prediction scheme.

**Why evaluate on this task?** This masking strategy presents a unique challenge to the model. By disrupting the local context with masked tokens, we force the model to rely on information from more distant, unmasked tokens to accurately predict the next word. This effectively evaluates the model's ability to capture and utilize longer range dependencies. A model with strong long-range capabilities will demonstrate lower perplexity on this task, indicating its proficiency in integrating information across extended sequences.

**Experimental results:** To evaluate MELODI's performance on this new task, we conducted experiments using the PG-19 dataset (with the T5 vocabulary) and two different architectures: 12-layer and 36-layer. We applied random masking with three masking ratios (0.125, 0.25, and 0.5) to systematically vary the degree of contextual disruption. The batch sequence length is 4096, comprising 8 context windows of 512 tokens.

As expected, perplexity increased across all models as the masking ratio increased, indicating the growing difficulty of the task. However, MELODI consistently outperformed both Transformer XL and the Memorizing Transformer across all masking ratios and architectural configurations. These results, presented in Table D.5, strongly suggest that MELODI exhibits superior capabilities in capturing and utilizing long-range dependencies compared to these baseline models.

Interestingly, even with a high masking ratio of 0.5, the 12-layer MELODI variant ($S_{192} + L_{96}$) achieves a perplexity of 52.59, which is remarkably close to the 36-layer Transformer XL's perplexity of 51.23. This suggests that MELODI can achieve comparable performance with significantly fewer parameters.

**Slower perplexity increase for MELODI with increasing mask ratio:** We further examined the change in perplexity as the masking ratio increases: $0\rightarrow0.125$, $0.125\rightarrow0.25$, and $0.25\rightarrow0.5$. As shown in Table D.5 (see numbers in the bracket), the perplexity increase for MELODI is consistently smaller than that observed for the baseline models. This indicates that MELODI is less affected by the increasing disruption of local context, further highlighting its superior ability to capture and leverage long-range dependencies.

### D.6 MORE ABLATIONS

In this section, we present additional ablation studies.

**Directly copying short-term memory to long-term memory:** This ablation experiment, conducted at the long-term layer, explores directly copying short-term memory tokens as long-term tokens. Instead of generating long-term tokens using the linear token mixer, this approach utilizes the short-term tokens present at the long-term layer and stores them directly in the long-term memory. Here, we force the short-term and long-term to share the same number of tokens ($S = L$). The results in Table 10 indicate that this direct copying method leads to a performance degradation.

**Effect of long-term layer position:** We investigate the impact of varying the position of the long-term layer ($M$ in Figure 1) within a 13-layer MELODI model ($S_{128}+L_{64}$). Layers are indexed from

Table 9: **Performance comparison on masked next token prediction with varying mask ratios.** This table presents the perplexity of Transformer XL, Memorizing Transformer, and MELODI on the PG-19 dataset (T5 vocabulary) across different mask ratios. Each column represents a different masking level, and each model is evaluated with both 12 and 36 layers. The number in the bracket indicates the perplexity increase from the previous column due to the increase of mask ratio.

| MODEL | #Layers | Mask Ratio | | | |
|---|---|---|---|---|---|
| | | no mask | 0.125 | 0.25 | 0.5 |
| Transformer XL | 12 | 11.54 | $16.08_{(+4.54)}$ | $22.50_{(+6.42)}$ | $65.51_{(+43.01)}$ |
| Memorizing Transformer | 12 | 10.74 | $14.91_{(+4.17)}$ | $20.76_{(+5.85)}$ | $56.28_{(+35.52)}$ |
| **MELODI** $S_{128} + L_{64}$ | 12 | 10.61 | $14.58_{(+3.97)}$ | $20.09_{(+5.51)}$ | $53.63_{(+33.54)}$ |
| **MELODI** $S_{192} + L_{96}$ | 12 | **10.48** | $\mathbf{14.39}_{(+3.91)}$ | $\mathbf{20.00}_{(+5.61)}$ | $\mathbf{52.59}_{(+32.59)}$ |
| Transformer XL | 36 | 9.61 | $13.10_{(+3.49)}$ | $18.18_{(+5.08)}$ | $51.23_{(+33.05)}$ |
| Memorizing Transformer | 36 | 8.92 | $12.15_{(+3.23)}$ | $16.64_{(+4.49)}$ | $46.80_{(+30.16)}$ |
| **MELODI** $S_{128} + L_{64}$ | 36 | 8.81 | $11.97_{(+3.16)}$ | $16.21_{(+4.24)}$ | $45.57_{(+29.36)}$ |
| **MELODI** $S_{192} + L_{96}$ | 36 | **8.70** | $\mathbf{11.74}_{(+3.04)}$ | $\mathbf{15.98}_{(+4.24)}$ | $\mathbf{44.05}_{(+28.07)}$ |

Table 10: **Directly copying short-term tokens to long-term memory:** Here, we force the short-term and long-term to share the same number of tokens (e.g. $S_{64}+L_{64}$), and examine the impact of directly using short-term tokens as long-term tokens, bypassing the linear token mixer. This approach results in performance degradation compared to generating distinct long-term tokens.

| Copying | $S_{96}+L_{96}$ | $S_{64}+L_{64}$ | $S_{32}+L_{32}$ |
|---|---|---|---|
| Yes | 11.00 | 11.36 | 11.42 |
| No | **10.92** | **11.08** | **11.25** |

Table 11: **Position of long-term layer.** Here, we use a 13-layer MELODI model, where layers are indexed from 0 to 12. The model uses 128 short-term and 64 long-term tokens per context window. While the default position of the long-term layer is at layer 8, placing it at different layers between 5 and 11 yields consistent perplexity scores.

| **Layer** | 5 | 6 | 7 | 8* |
|---|---|---|---|---|
| **Perplexity** | 11.00 | 11.01 | 11.03 | 10.95 |
| **Layer** | 9 | 10 | 11 | |
| **Perplexity** | 10.96 | 10.95 | 10.94 | |

0 to 12, with the default long-term layer position at layer 8. Results in Table 11 reveal consistent perplexity scores when the long-term layer is positioned between layers 5 and 11.

# E    ANALYSIS OF MEMORY USAGE AND COMPUTATIONAL COMPLEXITY

This section analyzes the memory usage and computational complexity of MELODI, including empirical measurements of its training and test times compared to the baselines.

## E.1    NOTATIONS

To facilitate our analysis, we first define the following notations:

- $N$: Number of layers in the model.
- $W$: Number of context tokens per window.
- $D$: Dimension of the token embeddings.
- $S$: Number of short-term memory tokens per window.
- $Q$: Number of windows covered by long-term memory in the Memorizing Transformer.
- $R$: MELODI's long-term memory reduction rate compared to the Memorizing Transformer.

Table 12: **Comparison of memory usage between MELODI and the baseline models**. Notations are introduced in Appendix E.1.

| MODEL | Short-Term Memory | Long-Tem Memory |
|---|---|---|
| Transformer XL | $2NWD$ | $0$ |
| Memorizing Transformer | $2NWD$ | $2QWD$ |
| **MELODI** | $NSD$ | $\frac{2QWD}{R}$ |

Table 13: **Memory usage and model size** for MELODI and baseline models. Memory usage is measured in number of floats and includes both short-term and long-term memory. Model size is reported in number of parameters.

| MODEL | #Layers | Embedding Dim | Model Size | Window Length | Memory Size | $\frac{\text{Memory Size}}{\text{Model Size}}$ |
|---|---|---|---|---|---|---|
| Transformer XL | 12 | 1024 | 151M | 512 | 12.6M | 0.08 |
| Memorizing Transformer | 12 | 1024 | 151M | 512 | 146.8M | 0.97 |
| **MELODI** $S_{128} + L_{64}$ | 12 | 1024 | 153M | 512 | 18.4M | 0.12 |
| Transformer XL | 16 | 1536 | 453M | 512 | 25.2M | 0.05 |
| Memorizing Transformer | 16 | 1536 | 453M | 512 | 226.5M | 0.50 |
| **MELODI** $S_{128} + L_{64}$ | 16 | 1536 | 456M | 512 | 28.3M | 0.06 |
| Transformer XL | 32 | 4096 | 6.7B | 4096 | 1.07B | 0.16 |
| Memorizing Transformer | 32 | 4096 | 6.7B | 4096 | 5.37B | 0.80 |
| **MELODI** $S_{1024} + L_{512}$ | 32 | 4096 | 6.8B | 4096 | 0.67B | 0.10 |

### E.2 MEMORY USAGE

Table E.1 compares the short-term and long-term memory size of MELODI and the baseline models. For example, the MELODI $S_{128} + L_{64}$ variant uses a quarter of the window length for short-term memory ($S = W/4$) and achieves a long-term memory reduction rate of $R = 8$, resulting in an $8\times$ reduction in memory usage compared to the Memorizing Transformer.

To further illustrate the relationship between memory footprint and model size, we compare three different model configurations in Table 13. This includes the two smaller Transformer models used in our experiments and the Llama-2 (Touvron et al., 2023) 7B model with 32 layers, an embedding dimension of 4096, and a context window of 4096. For this comparison, we set the long-term memory to cover 128 context windows. The table demonstrates that the memory size for the Memorizing Transformer is comparable to the model size itself. In contrast, MELODI's memory size remains significantly smaller—by an order of magnitude—even for a large model like Llama-2. This highlights the memory efficiency of MELODI's approach, especially for large-scale language models.

### E.3 COMPUTATIONAL COMPLEXITY

As shown in Table 14, the Memorizing Transformer introduces additional complexity compared to Transformer XL due to the cross-attention computation over the long-term memory in its middle layer. MELODI reduces this cross-attention complexity by a factor of $R$ compared to the Memorizing Transformer. However, it introduces additional computations in the query-key-value, feedforward network, and linear token mixing components. The extent of these additional computations is controlled by the number of short-term memory tokens, $S$, which is about a quarter of window length $W$.

Table 14: **Component-wise comparison of computational complexity between MELODI and the baseline models**. Notations are introduced in Appendix E.1.

| MODEL | Query- Key-Value | Self- Attention | Cross- Attention | Feedforward Network | Linear Token Mixing |
|---|---|---|---|---|---|
| Trans. XL | $\mathcal{O}(NWD^2)$ | $\mathcal{O}(NW^2D)$ | – | $\mathcal{O}(NWD^2)$ | – |
| Mem Trans. | $\mathcal{O}(NWD^2)$ | $\mathcal{O}(NW^2D)$ | $\mathcal{O}(QWD)$ | $\mathcal{O}(NWD^2)$ | – |
| **MELODI** | $\mathcal{O}(N(W{+}S)D^2)$ | $\mathcal{O}(NW^2D)$ | $\mathcal{O}(\frac{QWD}{R})$ | $\mathcal{O}(N(W{+}S)D^2)$ | $\mathcal{O}(N(W{+}S)SD)$ |

Table 15: **Comparison of training and test times for MELODI and baseline models.**. We measured the training and test times of 12-layer networks with an embedding dimension of 1024 on TPU v6e. Experiments were conducted with sequence lengths of 4K and 8K tokens and window lengths of 512 and 1K tokens.

| MODEL | Segment Length (num of tokens) | Window Length (num of tokens) | Training Time (sec) | Test Time (sec) |
|---|---|---|---|---|
| Transformer XL | 4096 | 512 | 0.052 | 0.022 |
| Memorizing Trans. | 4096 | 512 | 0.094 | 0.033 |
| **MELODI** $S_{128}{+}L_{64}$ | 4096 | 512 | 0.076 | 0.023 |
| **MELODI** $S_{192}{+}L_{96}$ | 4096 | 512 | 0.097 | 0.027 |
| Transformer XL | 4096 | 1024 | 0.068 | 0.027 |
| Memorizing Trans. | 4096 | 1024 | 0.108 | 0.036 |
| **MELODI** $S_{256}{+}L_{128}$ | 4096 | 1024 | 0.094 | 0.028 |
| **MELODI** $S_{384}{+}L_{192}$ | 4096 | 1024 | 0.111 | 0.031 |
| Transformer XL | 8192 | 1024 | 0.108 | 0.045 |
| Memorizing Trans. | 8192 | 1024 | 0.196 | 0.067 |
| **MELODI** $S_{256}{+}L_{128}$ | 8192 | 1024 | 0.164 | 0.047 |
| **MELODI** $S_{384}{+}L_{192}$ | 8192 | 1024 | 0.204 | 0.054 |

### E.4 TRAINING AND TEST TIMES

To evaluate the computational efficiency of MELODI, we measured its training and test time and compared it against the baselines. All experiments were conducted using 12-layer networks with an embedding dimension of 1024 on TPU v6e. We evaluated performance with batch sequence lengths of 4K and 8K tokens and window lengths of 512 and 1K tokens.

As shown in Table 15, MELODI consistently demonstrates faster ***test times*** than the Memorizing Transformer. Impressively, the MELODI $S_{128}{+}L_{64}$ variant approaches the test speed of Transformer XL (e.g., 0.022 vs. 0.023 seconds with a 4K sequence length and 512 window length).

However, when considering ***training time***, both MELODI variants are comparable to the Memorizing Transformer but slower than Transformer XL. This suggests that the additional computations introduced by MELODI's short-term memory component (in the query-key-value, feedforward network, and linear token mixing) are relatively inexpensive during test (forward pass) but become more costly during backpropagation.

