# OpenReview forum: "MELODI: Exploring Memory Compression for Long Contexts"
_ICLR.cc/2025/Conference — ICLR 2025 Poster_

### Official Review · Reviewer_i9z9 · 2024-11-02

**Soundness:** 3
**Presentation:** 3
**Contribution:** 2
**Rating:** 6
**Confidence:** 2

**Summary:**

This paper presents MELODI, a memory architecture for efficient long document processing. It utilizes a hierarchical compression scheme, where the short-term memory performs recurrent compression of context windows, and long-term memory aggregates information from the entire history. Experimental results indicate superior performance along with a reduction in memory footprint.

**Strengths:**

1. The writing is clear and fluent.
2. Experimental results are promising, demonstrating an improvement in performance while simultaneously reducing memory usage.

**Weaknesses:**

1. The baselines are old. Some related methods are not discussed or compared.
2. Some experimental details are not provided. For example, how to choose the M, and how to evaluate the memory footprint.
3. For Figure 8, with the number of long-term memory layers increasing, the perplexity is reduced. So a single layer may not be sufficient.

**Questions:**

1. Could you add some latest methods, e.g. [1], to related works and baselines?
2. Could you provide more details about the experiments, e.g. how to choose the M, and how to evaluate the memory footprint?
3. There is only one long-term layer in the proposed method, which may not be sufficient to model long-term memory. As shown in Figure 8, could you conduct experiments with more long-term memory layers?

[1] Wang, Weizhi, et al. "Augmenting language models with long-term memory." Advances in Neural Information Processing Systems 36 (2024).

---

> ### Author Response · Authors · 2024-11-20
> **Response from authors**
>
> We sincerely thank the reviewer for the thoughtful feedback and valuable suggestions, which have significantly improved the quality of our paper.
>
> $\color{blue}{\textbf{[Weakness 1 + Question 1]:}}$
>
> **The baselines are old. Some related methods are not discussed or compared. Could you add some latest methods, e.g. [1], to related works and baselines?**
>
> **[1] Wang, Weizhi, et al. "Augmenting language models with long-term memory." Advances in Neural Information Processing Systems 36 (2024).**
>
> Thank you for the suggestion. We agree that comparing MELODI with LONGMEM [1] would provide valuable insights. We will add LONGMEM to the related work section and discussed its relationship to MELODI.
>
> While both LONGMEM and MELODI improve upon the Memorizing Transformer, they do so with different focuses and approaches:
>
> * ***LONGMEM***: LONGMEM focuses on improving prediction accuracy in a fine-tuning setting by introducing a SideNet for memory retrieval and fusion. It is compared against the Memorizing Transformer variant that uses top-k KV pairs from memory.
>
> * ***MELODI***: MELODI prioritizes efficiency in a train-from-scratch setting by compressing both long-term and short-term memory. It introduces a negligible number of extra parameters and is compared against the Memorizing Transformer variant that uses all KV pairs in memory.
>
> A direct experimental comparison with LONGMEM in the fine-tuning (of LLMs) setting is currently out of the scope of this paper, as our primary focus is on evaluating MELODI's performance in the train-from-scratch setting. However, we plan to explore extending MELODI to the fine-tuning setting in our future work, where we will include a comprehensive comparison with LONGMEM and other relevant baselines. This will provide a more complete picture of MELODI's capabilities and its relative strengths compared to other memory-augmented language models.
>
> ---
> $\color{blue}{\textbf{[Weakness 2 + Question 2]:}}$
>
> **more details about the experiments, e.g. how to choose the $M$, and how to evaluate the memory footprint?**
>
> *Hyperparameter $M$:*
>
> $M$ in Figure 1 denotes the index of the long-term layer within the transformer architecture. For a 13-layer network, the default value is $M=8$. Ablation in Appendix D (Table 8) demonstrates that model performance is robust to variations in $M$. For convenience, we attach the table below.
>
>
> | Position of Long-term Layer $M$ | Perplexity $\downarrow$ |
> |:-:|:-:|
> | Layer 5 | 11.00 |
> | Layer 6 | 11.01 |
> | Layer 7 | 11.03 |
> | Layer 8 * | 10.95 |
> | Layer 9 | 10.96 |
> | Layer 10 | 10.95 |
> | Layer 11 | 10.94 |
>
> *Memory Footprint:*
>
> Memory size is measured as the total number of floating-point numbers (floats) used to store model activations. For example, *MELODI S128+L64* utilizes 128 short-term memory tokens per layer. In a 13-layer transformer with a token dimension of 1024, this results in 128 × 1024 × 13 = 1.7M floats for short-term memory. Additionally, the long-term memory stores 64 key-value pairs (128 tokens) per context window. To cover 128 context windows (64k context tokens if each window has 512 tokens), the long-term memory requires 64 × 2 × 1024 × 128 = 16.8M floats.
>
> ---
> $\color{blue}{\textbf{[Weakness 3 + Question 3]:}}$
>
> **Only one long-term layer in the proposed method may not be sufficient to model long-term memory. Could you conduct experiments with more long-term memory layers in Figure 8?**
>
>
> Excellent point. We conducted additional ablations to extend the number of long-term layers to 6 and report the results in the Table below. The perplexity plateaus when using more than 3 long-term layers. Compared to the maximum improvement of 0.60 (from 11.39 with no long-term layer to 10.79 with 6 layers), the first long-term layer alone contributes 73% (0.44/0.60) of this gain.  Since additional long-term layers increase both memory footprint and computational cost due to cross-attention, a single long-term layer offers a good balance between performance and efficiency.
>
> | Num of Long-Term Layer | Perplexity $\downarrow$ |
> |:-:|:--|
> | 0 | 11.39 |
> | 1 | 10.95 $_{(-0.44)}$|
> | 2 | 10.84 $_{(-0.55)}$|
> | 3 | 10.79 $_{(-0.60)}$|
> | 4 | 10.81 $_{(-0.58)}$|
> | 5 | 10.80 $_{(-0.59)}$|
> | 6 | 10.79 $_{(-0.60)}$|

---

> > ### Author Response · Authors · 2024-11-22
> > **Reminder: Author Response and Follow Up**
> >
> > Dear Reviewer i9z9,
> >
> > We hope this message finds you well. We have posted our responses to your valuable feedback and appreciate you taking the time to review them.  We understand the demands on your time and deeply appreciate your active participation in this discussion.
> >
> > As the author-reviewer discussion period nears its close, we want to reiterate that we are available to answer any further questions or concerns you may have about our rebuttal. Your continued insights are crucial in helping us further refine our manuscript.
> >
> > Please don't hesitate to reach out if you have any further thoughts or inquiries.
> >
> > Sincerely,
> >
> > The Authors

---

> > > ### Author Response · Authors · 2024-11-25
> > > **Re: Author Response and Follow Up**
> > >
> > > Dear Reviewer i9z9,
> > >
> > > We wanted to follow up on our previous message and see if you had any further thoughts or questions about our response to your valuable feedback.  We understand how busy you are, and we truly appreciate you taking the time to engage with our work.
> > >
> > > The author-reviewer discussion period is closing in two days. We are very grateful for your insights and would be happy to clarify any aspects of our rebuttal or address any remaining concerns.
> > >
> > > Please don't hesitate to reach out if you have any further questions.
> > >
> > > Sincerely,
> > >
> > > The Authors

---

> > > > ### Comment · Reviewer_i9z9 · 2024-11-26
> > > >
> > > > Thanks for the detailed response, which addressed most of my concerns. Based on the response to question 2, "Memory size is measured as the total number of floating-point numbers (floats) used to store model activations, " so the memory footprint is just for the memory tokens, not the whole model. How does this memory footprint compare to the model's parameters? Specifically, if this method is applied to large models, such as Llama-2, the memory footprint for the memory tokens may occupy a small portion of the total memory footprint.

---

> ### Author Response · Authors · 2024-11-26
> **Memory Footprint Relative to Model Size**
>
> Thanks for the follow-up question. We agree that it's important to compare the memory footprint of the memory tokens to the overall model size, especially when considering larger models. To facilitate this analysis, we first define the following notation:
>
> $N$: Number of layers in the model
>
> $W$: Number of context tokens per window
>
> $D$: Dimension of the token embeddings
>
> $S$: Number of short-term memory tokens per window
>
> $Q$: Number of windows covered by long-term memory in Memorizing Transformer
>
> $R$: MELODI's Long-term memory redunction rate compared to Memorizing Transformer
>
> The table below compares the short-term and long-term memory size of MELODI and the baseline models.
>
> | Method | Short-term Memory | Long-term Memory |
> |---|:-:|:-:|
> | Transformer XL | $2NWD$| 0|
> | Memorizing Transformer | $2NWD$| $2QWD$ |
> | **MELODI** | $NSD$  | $\frac{2QWD}{R}$|
>
> For example, the MELODI S128+L64 variant uses a quarter of the window length for short-term memory ($S=W/4$) and achieves a long-term memory reduction rate of $R=8$. This results in an 8x reduction in memory usage compared to the Memorizing Transformer.
>
> To further illustrate the relationship between memory footprint and model size, we present a comparison for three different model configurations in the table below. The first two are the smaller Transformer models, while the third is the Llama-2 7B model with 32 layers, an embedding dimension of 4096, and a context window of 4096.  For this comparison, we set the long-term memory to cover 128 context windows.
>
>
> | Method | #Layers | Embedding Dim | Model Size | Window Length | Memory Size | $\frac{\text{memory size}}{\text{model size}}$
> |---|:-:|:-:|:-:|:-:|:-:|:-:|
> | Transformer XL | 12 | 1024 | 151M | 512 | 12.6M | 0.08 |
> | Memorizing Transformer | 12 | 1024 | 151M | 512 | 146.8M | 0.97 |
> | **MELODI S128+L64** | 12 | 1024 | 153M | 512 | 18.4M | 0.12 |
> ||
> | Transformer XL | 16 | 1536 | 453M | 512 | 25.2M | 0.05 |
> | Memorizing Transformer | 16 | 1536 | 453M | 512 | 226.5M | 0.50 |
> | **MELODI S128+L64** | 16 | 1536 | 456M | 512 | 28.3M | 0.06 |
> ||
> | Transformer XL | 32 | 4096 | 6.7B | 4096 | 1.07B | 0.16 |
> | Memorizing Transformer | 32 | 4096 | 6.7B | 4096 | 5.37B | 0.80 |
> | **MELODI S1024+L512** | 32 | 4096 | 6.8B | 4096 | 0.67B | 0.10 |
>
> As you can see, the memory size for the Memorizing Transformer is comparable to the model size, while MELODI's memory size remains significantly smaller—by an order of magnitude—even for a large model like Llama-2. We will add this analysis to the revised manuscript to provide a clearer comparison of memory footprint relative to model size.

---

> > ### Author Response · Authors · 2024-11-27
> > **Revised Manuscript with Incorporated Feedback**
> >
> > Dear Reviewer i9z9,
> >
> > Following up on our previous message, we wanted to let you know that we have carefully revised our manuscript based on your valuable feedback, including your recent comments on the comparison between memory footprint and model size (see the list below).
> >
> > The revised manuscript is available for your review. We believe that these changes strengthened the paper.
> >
> > The author-reviewer discussion period is closing soon. If you have any further questions or require any clarifications, please don't hesitate to reach out. We appreciate your time and consideration.
> >
> > Sincerely,
> >
> > Authors
> >
> > | Description | Location in the revised manuscript | Related Weakness/Question |
> > |:--|:--|:--|
> > |Comparison with LongMem | Line 100 in Section 2 (Related Work), Line 791-805 in Appendix B | Weakness 1, Question 1 |
> > |Ablation on long-term layer position $M$ | Line 1105-1108 + Table 11 in Appendix D.6 | Weakness 2, Question 2 |
> > |Memory measured as the number of floats | Line 311-317 in Section 3.3 | Weakness 2, Question 2 |
> > |Experiments with more long-term memory layers in Figure 8| Line 517-523 + updated Figure 8 in Section 4.4 | Weakness 3, Question 3 |
> > |Memory Size vs. Model Size | Line 1127-1167 + Table 12, 13 in Appendix E.2 | Followup |

---

### Official Review · Reviewer_hBrU · 2024-11-03

**Soundness:** 3
**Presentation:** 3
**Contribution:** 3
**Rating:** 6
**Confidence:** 3

**Summary:**

The paper introduces a novel Transformer-based architecture, MELODI, which combines multiple short-term context window (recurrent + chunk-wise) Transformer layers with a single long-term compression layer. Experimental results show that MELODI achieves the lowest perplexity compared to similar memory-based Transformer variants. In addition, the paper presents ablation studies exploring various combinations of architectural components.

**Strengths:**

* The integration of long and short memories is a straightforward yet effective strategy. The paper clearly illustrates the concept through well-written explanations and clear visualizations.

**Weaknesses:**

* The experiments mainly focus on perplexity comparisons, rather than evaluating the performance on long-context benchmarks such as LongBench or Needle-in-a-Haystack (NIH).
* The paper lacks discussion regarding the challenges in training the model; the recurrent nature of the model may impede efficient parallel training. Can you provide more detailed explanations?
* Although the model is Decoder-only, it is unclear how the chunk-wise approach affects token-wise generation behavior (a key mechanism in GPT-like LLMs). Does summary-generating action occur only after completing each chunk during the generation phase?
* The model appears to be relatively small, raising questions about the scalability of the architecture. Details regarding the model size (e.g., 1B, 2B?) and the model’s potential for scaling up would enhance the paper’s value.

**Questions:**

* It would be beneficial to provide the upper bound, such as a very long-context Transformer model (ex: 256K or more) for readers to get a sense of the performance gap.
* Suggestion: It would be interesting to compare MELODI with YOCO [1], which employs a similar long-short strategy. I understand that [1] is a very recent paper and you do not need to compare the work; it would be valuable to share your thoughts on the similarities and differences. [1] You Only Cache Once: Decoder-Decoder Architectures for Language Models, 2024.

---

> ### Author Response · Authors · 2024-11-20
> **Response from authors - part 1**
>
> We sincerely thank the reviewer for the thoughtful feedback and valuable suggestions, which have significantly improved the quality of our paper.
>
> $\color{blue}{\textbf{[Weakness 1]:}}$
>
> **The experiments mainly focus on perplexity comparisons, rather than evaluating the performance on long-context benchmarks such as LongBench or Needle-in-a-Haystack (NIH).**
>
> We acknowledge the limitation of not evaluating MELODI on benchmarks such as LongBench or Needle-in-a-Haystack. This will involve scaling up the model and incorporating MELODI into a pre-train LLMs using fine-tuning or LoRA, which we plan to explore in future work.  (Please see our response to other reviewers.)
>
> Although a comprehensive evaluation on downstream tasks is impractical within the rebuttal period, we introduce a novel task designed to further evaluate MELODI's capacity for handling long-range dependencies.
>
> *New Task: Masked Next Token Prediciton*
>
> Similar to our primary evaluation, we use next token prediction and measure performance with perplexity. However, we introduce a key challenge: masking out a portion of the input tokens. Unlike BERT's bidirectional masking approach, this new task employs a unidirectional, left-to-right prediction scheme.
>
> *Why evaluate on this task?*
>
> This masking strategy presents a unique challenge to the model. By disrupting the local context with masked tokens, we force the model to rely on information from more distant, unmasked tokens to accurately predict the next word. This effectively evaluates the model's ability to capture and utilize longer range dependencies. A model with strong long-range capabilities will demonstrate lower perplexity on this task, indicating its proficiency in integrating information across extended sequences.
>
> *Experimental Results*
>
> To evaluate MELODI's performance on this new task, we conducted experiments using the PG-19 dataset (with the T5 vocabulary) and two different architectures: 12-layer and 36-layer. We applied random masking with three masking ratios (0.125, 0.25, and 0.5) to systematically vary the degree of contextual disruption.
>
> As expected, perplexity increased across all models as the masking ratio increased, indicating the growing difficulty of the task. However, MELODI consistently outperformed both TransformerXL and the Memorizing Transformer across all masking ratios and architectural configurations. These results, presented in the table below, strongly suggest that MELODI exhibits superior capabilities in capturing and utilizing long-range dependencies compared to these baseline models.
>
> | Method | Mask Ratio | Num of Layers |Perplexity $\downarrow$ | | Num of Layers |Perplexity $\downarrow$ |
> |---|:-:|:-:|:-:|---|:-:|---|
> |Transformer XL| no mask | 12 | 11.54 | | 36 | 9.61 |
> |Memorizing Transformer| no mask | 12 | 10.74 | | 36 | 8.92 |
> |**MELODI S128+L64**| no mask | 12 | 10.61 | | 36 | 8.81 |
> |**MELODI S192+L96**| no mask | 12 | **10.48** || 36 | **8.70** |
> ||
> |Transformer XL| 0.125 | 12 | 16.08 | | 36 | 13.10 |
> |Memorizing Transformer| 0.125 | 12 | 14.91 | | 36 | 12.15 |
> |**MELODI S128+L64**| 0.125 | 12 | 14.58 | | 36 | 11.97 |
> |**MELODI S192+L96**| 0.125 | 12 | **14.39** || 36 | **11.74** |
> ||
> |Transformer XL| 0.25 | 12 | 22.50 | | 36 | 18.18 |
> |Memorizing Transformer| 0.25 | 12 | 20.76 | | 36 | 16.64 |
> |**MELODI S128+L64**| 0.25 | 12 | 20.09 | | 36 | 16.21 |
> |**MELODI S192+L96**| 0.25 | 12 | **20.00** || 36 | **15.98** |
> ||
> |Transformer XL| 0.5 | 12 | 65.51 | | 36 | 51.23 |
> |Memorizing Transformer| 0.5 | 12 | 56.28 | | 36 | 46.80 |
> |**MELODI S128+L64**| 0.5 | 12 | 53.63 | | 36 | 45.57 |
> |**MELODI S192+L96**| 0.5 | 12 | **52.59** || 36 | **44.05** |

---

> ### Author Response · Authors · 2024-11-20
> **Response from authors - part 2**
>
> *Slower Perplexity Increase for MELODI with Increasing Mask Ratio*
>
> Interestingly, even with a high masking ratio of 0.5, the 12-layer MELODI variant (S192+L96) achieves a perplexity of 52.59, which is remarkably close to the 36-layer TransformerXL's perplexity of 51.23. This suggests that MELODI can achieve comparable performance with significantly fewer parameters.
>
> To further analyze this trend, we examined the change in perplexity as the masking ratio increases: 0 → 0.125, 0.125 → 0.25, and 0.25 → 0.5. As shown in the table below, the perplexity increase for MELODI is consistently smaller than that observed for the baseline models. This indicates that MELODI is less affected by the increasing disruption of local context, further highlighting its superior ability to capture and leverage long-range dependencies.
>
> | Method | Num of Layers | Mask Ratio 0 $\rightarrow$ 0.125 | Mask Ratio 0.125 $\rightarrow$ 0.25 | Mask Ratio 0.25 $\rightarrow$ 0.5 |
> |---|:-:|:-:|:-:|:-:|
> | Transformer XL | 12 | +4.54 | +6.42 | +43.01 |
> | Memorizing Transformer | 12 | +4.17 | +5.85 | +35.52 |
> | **MELODI S128+L64** | 12 | +3.97 | +5.51 | +33.54 |
> | **MELODI S192+L96** | 12 | +3.91 | +5.61 | +32.59 |
> ||
> | Transformer XL | 36 | +3.49 | +5.08 | +33.05 |
> | Memorizing Transformer | 36 | +3.23 | +4.49 | +30.16 |
> | **MELODI S128+L64** | 36 | +3.16 | +4.24 | +29.36 |
> | **MELODI S192+L96** | 36 | +3.04 | +4.24 | +28.07 |
>
> ---
> $\color{blue}{\textbf{[Weakness 2]:}}$
>
> **The paper lacks discussion regarding the challenges in training the model; the recurrent nature of the model may impede efficient parallel training. Can you provide more detailed explanations?**
>
> Thank you for the suggestion. We will add the following details about efficient parallel training, addressing the challenges posed by the model's recurrent nature.
>
> *Batch Processing Strategy*
>
> To enable efficient parallel training despite the recurrent dependencies in the short-term and long-term memory, we employ a combined layer-wise and chunk-wise processing strategy.
>
> 1.  ***Layer-wise Processing:***  As is standard practice, MELODI uses a Transformer architecture with multiple layers, and the layers are processed in sequence, with the output of each layer becoming the input to the next.  Both short-term and long-term memory operations are confined within their respective layers.
>
> 2. ***Unrolling Multiple Windows***:  Within a layer, each training batch covers 8 context windows (512 tokens per window) with a total of 4096 tokens, allowing for backpropagation through time (BPTT) across multiple windows.  Although the windows must be processed sequentially due to window (i.e. block)  level recurrence, self-attention within each window can be done in parallel.  We achieve good device utilization so long as the window size is large enough.
>
> 3. ***Parallel and Sequential Operations:*** The computation of query, key, and value matrices (of context and summary tokens) for attention are computed in parallel across all chunks, as they do not depend on the recurrent memory updates. Therefore, each layer involves two steps: (a) parallel computation of query/key/value matrices, and (b) sequential computation of self-attention, feed-forward networks, and token mixing for memory updates.
>
> This strategy differs from a chunk-wise then layer-wise approach, where each chunk must be processed by all layers for the entire network before the next chunk can be processed.
>
> ---
>
> $\color{blue}{\textbf{[Weakness 3]:}}$
>
> **It is unclear how the chunk-wise approach affects token-wise generation behavior (a key mechanism in GPT-like LLMs). Does summary-generating action occur only after completing each chunk during the generation phase?**
>
> Excellent point! MELODI's chunk-wise approach operates in two steps during generation:
>
> - *Token-wise Generation*: Within each chunk, MELODI generates tokens sequentially, similar to GPT-like LLMs, utilizing both short-term and long-term memory.
>
> - *Summary Generation*: After completing a chunk, MELODI generates all summary tokens in parallel, and updates short-term and long-term memory accordingly.
>
> This ensures that token-wise generation within a chunk remains unaffected by the chunk boundaries, preserving the key mechanism of GPT-like LLMs.

---

> ### Author Response · Authors · 2024-11-20
> **Response from authors - part 3**
>
> $\color{blue}{\textbf{[Weakness 4]:}}$
>
> **The model appears to be relatively small, raising questions about the scalability of the architecture. Details regarding the model size (e.g., 1B, 2B?) and the model’s potential for scaling up would enhance the paper’s value.**
>
> Thank you for the feedback. We acknowledge that the models evaluated in the main paper (12-layer Transformers with an embedding dimension of 1024, approximately 150M parameters) were relatively small. To address concerns about scalability, we conducted additional experiments on the PG-19 dataset (using the T5 vocabulary) with larger architectures. We explored two scaling approaches:
>
> 1. *Increased Depth*: We trained models with 24 and 36 layers, effectively scaling the number of parameters by 2 and 3 times, respectively.
>
> 2. *Increased Width and Depth*: We trained a model with 16 layers and an embedding dimension of 1536, which also scales the parameter count by 3 times.
>
> The results, presented in the table below, demonstrate that MELODI's advantages hold even with larger models. It consistently outperforms the Memorizing Transformer while using significantly less memory (approximately 8 times less).
>
> | Method | #Layers | Embedding Dim | Model Size | Memory Size | Perplexity $\downarrow$ |
> |---|:-:|:-:|:-:|--:|:-:|
> | Transformer XL | 12 | 1024 | 151M | 12.6M | 11.54 |
> | Memorizing Transformer | 12 | 1024 | 151M | 146.8M | 10.74 |
> | **MELODI S128+L64** | 12 | 1024 | 153M | 18.4M | 10.61 |
> | **MELODI S192+L96** | 12 | 1024 | 154M | 27.6M | **10.48** |
> ||
> | Transformer XL | 24 | 1024 | 302M | 25.2M | 10.15 |
> | Memorizing Transformer | 24 | 1024 | 302M | 159.4M | 9.45 |
> | **MELODI S128+L64** | 24 | 1024 | 306M | 20.0M | 9.30 |
> | **MELODI S192+L96** | 24 | 1024 | 309M | 30.0M | **9.16** |
> ||
> | Transformer XL | 36 | 1024 | 453M | 37.8M | 9.61 |
> | Memorizing Transformer | 36 | 1024 | 453M | 172.0M | 8.92|
> | **MELODI S128+L64** | 36 | 1024 | 459M | 21.6M | 8.81 |
> | **MELODI S192+L96** | 36 | 1024 | 463M | 32.4M | **8.70** |
> ||
> | Transformer XL | 16 | 1536 | 453M | 25.2M | 9.69 |
> | Memorizing Transformer | 16 | 1536 | 453M | 226.5M | 9.01|
> | **MELODI S128+L64** | 16 | 1536 | 456M | 28.3M | 8.89 |
> | **MELODI S192+L96** | 16 | 1536 | 459M | 42.5M | **8.79** |
>
> ---
>
> $\color{blue}{\textbf{[Question 1]:}}$
>
> **It would be beneficial to provide the upper bound, such as a very long-context Transformer model (ex: 256K or more) for readers to get a sense of the performance gap.**
>
> Thank you for the suggestion. Due to limited computational resources, we are unable to train a conventional Transformer model with a context window of 256K tokens or more. The maximum context window length we can achieve for a 12-layer transformer with an embedding size of 1024 is 4K tokens.
>
> However, we do provide a comparison against Memorizing Transformer, which does have a long context of 64k, and uses full dense attention.  Memorizing transformer only has one long-context layer, but prior work has demonstrated that there is limited benefit to adding multiple long-context layers.  At this scale, a single long-range layer seems to be sufficient.  (Our experiments confirm this finding, see response to reviewer i9z9 below.)  Going beyond 64k on PG19 also has a very limited benefit, since the average book length is only 69k.  Thus, the Memorizing Transformer baseline is a reasonable approximation of a conventional long-context model.
>
> To provide a further reference point for longer context windows using a conventional architecture, in which every layer has the same context length, we compare MELODI (with a context window of 512 tokens) to Transformer XL with context windows ranging from 512 to 4K tokens. All models are trained on the PG-19 dataset with the T5 vocabulary for 500k steps, with each batch including 8K tokens to allow backpropagation through time (BPTT) across multiple windows.
>
> As shown in the table below, Transformer XL's performance improves with increasing context length. Notably, MELODI (512 tokens) outperforms Transformer XL even with an 8 times longer context window (4K tokens).
>
> | Method | Batch Sequence Length | Context Window Length | Perplexity $\downarrow$ |
> |---|:-:|:-:|:-:|
> | Transformer XL | 8K | 512 | 11.40 |
> | Transformer XL | 8K | 1K | 11.18 |
> | Transformer XL | 8K | 2K | 10.94 |
> | Transformer XL | 8K | 4K | 10.65 |
> | **MELODI S192+L96** | 8K | 512 | **10.22** |

---

> > ### Author Response · Authors · 2024-11-20
> > **Response from authors - part 4**
> >
> > $\color{blue}{\textbf{[Question 2]:}}$
> >
> > **Suggestion: It would be interesting to compare MELODI with YOCO [1], which employs a similar long-short strategy. I understand that [1] is a very recent paper and you do not need to compare the work; it would be valuable to share your thoughts on the similarities and differences.**
> >
> > **[1] You Only Cache Once: Decoder-Decoder Architectures for Language Models, 2024.**
> >
> > Thank you for the insightful suggestion. We will gladly add YOCO [1] to the related work section and discuss its relationship to MELODI.
> >
> > Both YOCO and MELODI utilize a long-short memory strategy.  Their short-term memory components involve multiple layers, and both cache long-term information in a middle layer. However, they differ significantly in their specific design choices for these memory components, as detailed below.  Furthermore, they exhibit distinct properties in their long-term key-value (KV) memory:
> >
> > * ***YOCO***: YOCO employs sliding-window attention across multiple self-decoder layers for short-term memory and a KV cache in the middle layer for long-term memory. Importantly, YOCO demonstrates that this long-term KV cache is reusable for the latter half of the network, significantly improving pre-filling efficiency by enabling early exit.
> >
> > * ***MELODI***: MELODI implements a different long-short strategy based on compression. Short-term memory is achieved through recurrent compression of context windows across multiple layers, while long-term memory leverages further compression within a single middle layer.  MELODI demonstrates that long-term KV memory is compressible, significantly reducing its size.
> >
> > YOCO and MELODI enrich the design space for long-short memory models and offer complementary approaches. Combining their strengths could lead to even more effective models. For instance, exploring whether long-term memory can be both reusable (as in YOCO) and compressible (as in MELODI) is a promising avenue for future research. This could enable efficient caching with a compact KV memory footprint.

---

> > > ### Comment · Reviewer_hBrU · 2024-11-22
> > > **Thank you for the rebuttal**
> > >
> > > Thank you for the detailed response.
> > > Most of my concerns, except perplexity, are addressed.
> > >
> > > I understand that perplexity measured from the masked input can approximately represent the language modeling capability. However, perplexity would not reflect the performance in the actual scenarios [2, 3], and the gap is somewhat emphasized for smaller models [1].
> > >
> > > [1] Generating Benchmarks for Factuality Evaluation of Language Models
> > >
> > > [2] Can Perplexity Reflect Large Language Model's Ability in Long Text Understanding?
> > >
> > > [3] What is Wrong with Perplexity for Long-context Language Modeling?
> > >
> > > I am raising my score from 5 to 6, but I suggest adding more benchmarks.

---

> > > > ### Author Response · Authors · 2024-11-22
> > > > **Thank you for the continued feedback**
> > > >
> > > > Thank you for the follow-up comments and suggestions. We are grateful that you have raised your score after considering our response and appreciate you raising the important point about the limitations of perplexity, particularly its potential disconnect from real-world performance and other long-context benchmarks [1, 2, 3].  We will incorporate these references into our paper and discuss this gap in the limitations section.
> > > >
> > > > To address this in future work, we plan to integrate MELODI into pre-trained language models using efficient fine-tuning techniques like LoRA (Low-Rank Adaptation). This will enable a comprehensive evaluation of MELODI's effectiveness on a broader range of long-context benchmarks and facilitate direct comparisons with state-of-the-art baselines. We will also include this direction in the future work section of our paper.

---

### Official Review · Reviewer_XNtf · 2024-11-03

**Soundness:** 3
**Presentation:** 3
**Contribution:** 3
**Rating:** 8
**Confidence:** 4

**Summary:**

MELODI is a novel memory architecture designed to efficiently process long documents using short context windows. MELODI is a hierarchical compression scheme for both short-term and long-term memory.

The Short-Term Memory uses a Multi-Layer Recurrent Compression Scheme which is distributed across multiple short-term layers, this mechanism progressively compresses context tokens and prior memory at each layer. It is asserted that this ensures smooth transitions between context windows and aggregates information across them. Each short-term layer: (1) Transforms context tokens via a transformer block and (2) Recurrently compresses the current context window into short-term memory *tokens*. Summary tokens are introduced to facilitate inter-layer communication within the short-term memory. Distinct linear token mixers are used to generate separate summary tokens for the next layer and short-term memory tokens for the next window, enabling divergent compression flows.

The Long-Term Memory is a Single-Layer Module which Memorizes Compressed Key-Value Pairs. It uses a single network layer and maintains a record of the entire history by further compressing each context window and stacking them. The claim is that this addresses the forgetting issue inherent in short-term memory due to limited capacity. Key-value (KV) pairs of long-term tokens are stored in a FIFO queue, ensuring the retention of a substantial portion of the recent history. The long-term layer introduces a long-term memory component and enables cross-attention between current context/summary tokens and the long-term memory. It employs a gating mechanism to combine self-attention and cross-attention results. It uses a linear token mixer to generate compressed KV pairs for the current window, which are then appended to the long-term memory.

The overall MELODI Architecture utilizes a "sandwich" structure with a long-term compression layer inserted between multiple recurrent short-term compression layers. It appears to demonstrate strong performance on various long-context datasets while significantly reducing memory usage compared to baselines like the Memorizing Transformer. Ablation studies appear to confirm the complementary roles of short-term and long-term memory, highlighting their synergistic contribution to an efficient and effective memory architecture.

Compared to LSTMs, which operate at the token level, MELODI focuses on memory management at the context window level. Unlike Transformer XL, which relies on caching KV pairs from the preceding window, MELODI employs a multi-layer recurrent compression for short-term memory. While similar to the Memorizing Transformer in using a dedicated memory layer, MELODI stores compressed KV pairs, leading to a substantial reduction in memory size.

**Strengths:**

Originality
This paper combines and extends ideas from previous works on memory mechanisms for language models. It integrates both short-term and long-term memory into a unified architecture, leveraging compression techniques inspired by LSTMs, Transformer XL, and Memorizing Transformer. The paper introduces a new hierarchical compression scheme for representing short-term and long-term memory. This scheme differentiates MELODI from previous approaches by employing recurrent compression across multiple layers for short-term memory and further compression within a single middle layer for long-term memory. The paper introduces a summary branching mechanism that uses linear token mixers to generate separate summary tokens for different purposes (next layer and next window). This contributes to the originality of the approach by enabling distinct compression flows within the short-term memory.

Quality
The paper presents comprehensive experimental results on three long-context datasets, demonstrating that MELODI outperforms several strong baselines (Transformer XL, Block Recurrent Transformer, and Memorizing Transformer) in terms of perplexity while significantly reducing memory usage. The paper conducts detailed ablation studies to analyze the impact of various design choices, including the sizes of short-term and long-term memory, long-term memory coverage, context window size, the number of short-term and long-term layers, and summary branching. These studies provide good evidence supporting the effectiveness and efficiency of MELODI's architectural choices.

Clarity
The paper is well-written and easy to follow. The authors present the motivation, architecture, and experimental results in a clear and concise manner. The paper uses several figures to effectively illustrate the key concepts and architectural components of MELODI. These figures significantly aid in understanding the hierarchical compression scheme and the flow of information within the model.  The authors provide detailed explanations of the various components of MELODI, such as the short-term and long-term layers, summary branching, and the gating mechanism used in the long-term layer.

Significance
The paper tackles the important problem of efficiently processing long documents with transformer language models. This is a significant challenge because the quadratic complexity of attention mechanisms makes it computationally expensive to handle long sequences directly.
The proposed method has practical implications for various real-world applications that involve processing long texts, such as document summarization, question answering, and machine translation. MELODI's efficiency and effectiveness in handling long contexts could lead to improved performance and reduced computational costs in these areas. The paper does point to some new avenues for future research in memory architectures for language models. The hierarchical compression scheme and the use of summary branching could inspire further investigations into efficient and effective ways to manage information over extended sequences. Additionally, the authors acknowledge the limitation of their current work in addressing the fine-tuning of pre-trained models and suggest exploring adaptation techniques like LoRA fine-tuning in future work.

**Weaknesses:**

(1) The paper has only a limited exploration of longer context window size
The primary focus and the majority of the experiments revolve around a context window size of 512 tokens. This emphasis on relatively short context windows might limit the applicability of the method to tasks requiring significantly longer contexts, such as those involving very long documents or those benefiting from processing larger chunks of information simultaneously. The paper could be strengthened by conducting more extensive experiments with larger context window sizes, particularly those exceeding 2048 tokens or, ideally experiments in the 10k token regime. This would provide a better understanding of MELODI's performance and scalability in handling truly long contexts. Exploring how the model's memory capacity and compression mechanisms need to be adjusted for optimal performance with varying context lengths would be valuable. Figure 5 could be highlighted, discussed and presented more clearly and explicitly in terms of the long context issue. It seems this experiment does give perplexity numbers for 192x64=12k token context windows, but it is a bit hidden amongst the rest of the experiments given the importance of these longer regimes.

(2) There appears to be a lack of fine-tuning experiments
MELODI and the baselines, seem to have been trained from scratch. The paper doesn't explore the performance of MELODI when fine-tuned on a pre-trained language model. While most LLMs are derived from a larger pre-trained model, this is only a minor limitation given the amount of resources needed to train a SOTA grade LLM.  Investigating the adaptability of MELODI to be applied in fine-tuning scenarios would be beneficial. This could involve exploring techniques like LoRA (Low-Rank Adaptation) to integrate the short-term and long-term memory mechanisms into pre-trained models without significantly increasing the number of trainable parameters. Evaluating the performance of fine-tuned MELODI models on downstream tasks would provide insights into its effectiveness in practical applications. This is mentioned as possible future work.

(3) Limited Analysis of Computational Efficiency
More analysis and discussion of Memory Usage and Computational Time or computational complexity would be helpful. The paper could potentially benefit from discussions and comparisons of training and inference times for MELODI and the baselines

(4) Experiments are almost exclusively performed providing test set perplexity results. While there are many downstream tasks that could benefit from models yielding better perplexity, it would be more compelling to see the impact on other kinds of tasks and metrics where long range information is important.

**Questions:**

Suggestion: Could you provide some more experimental results in the 2k and larger token context regime?
Questions: Could you elaborate on any particular challenges and potential solutions for scaling MELODI to handle even larger context windows, such as 4096 or 8192 tokens, which are becoming increasingly common in language modelling?
Is the key issue "just" the size of the GPUs / TPUs that you/one has access to, or could you provide more (concise) insights on this point?

Could you provide a discussion and/or analysis of MELODI's computational efficiency, including complexity analysis or measurements and comparisons of training and inference times compared to baseline models? Latency is an issue in real world situations and it would be useful for the reader to understand the inference time implications here.

The paper highlights the potential of MELODI for various natural language processing applications, such as document summarization, question answering, and machine translation. Have you considered conducting preliminary experiments on any of these downstream tasks to demonstrate the practical benefits of MELODI in specific applications?  With those kinds of experiments and answers to the questions above I could potentially score this paper higher.

---

> ### Author Response · Authors · 2024-11-20
> **Response from authors - part 1**
>
> We sincerely thank the reviewer for the insightful comments and detailed explanations, which have significantly helped us improve the quality of this paper.
>
> $\color{blue}{\textbf{[Weakness 1.1 + Question 1]:}}$
>
> **The paper has only a limited exploration of longer context window size. The primary focus and the majority of the experiments revolve around a context window size of 512 tokens. ... ... Could you provide more experimental results in the 2k and larger token context regime? Could you elaborate on any particular challenges and potential solutions for scaling MELODI to handle even larger context windows, such as 4096 or 8192 tokens? Is the key issue "just" the size of the GPUs / TPUs that you/one has access to, or could you provide more (concise) insights on this point?**
>
> *Experiments with Longer Context Window*
>
> Thank you for the suggestion. To explore the performance with larger context windows, we conducted additional experiments with batch sequence lengths of 4K and 8K tokens and window lengths up to 2K tokens.  All models were trained on PG-19 (T5 vocabulary) with 12-layer architectures.
>
> As shown in the table below, MELODI continues to outperform the Memorizing Transformer with significantly less memory (approximately 8 times), even with larger context windows. Due to limited computational resources, we were unable to train MELODI with context windows exceeding 2K tokens.
>
>
> | Method | Sequence Length | Window Length | Memory Size | Perplexity $\downarrow$ |
> |---|:-:|:-:|--:|:-:|
> |Transformer XL| 4K | 512 | 12.6M | 11.54 |
> |Memorizing Transformer| 4K | 512 | 146.8M| 10.74 |
> |**MELODI S128+L64**| 4K | 512 | 18.4M | 10.61 |
> |**MELODI S192+L96**| 4K | 512 | 27.6M | **10.48** |
> ||
> |Transformer XL| 4K | 1K | 25.2M | 11.26 |
> |Memorizing Transformer| 4K | 1K | 159.4M | 10.64 |
> |**MELODI S256+L128**| 4K | 1K | 19.9M | 10.47 |
> |**MELODI S384+L192**| 4K | 1K | 29.9M | **10.36** |
> ||
> |Transformer XL| 8K | 1K | 25.2M | 11.18 |
> |Memorizing Transformer| 8K | 1K | 159.4M | 10.42 |
> |**MELODI S256+L128**| 8K | 1K | 19.9M | 10.27 |
> |**MELODI S384+L192**| 8K | 1K | 29.9M| **10.19** |
> ||
> |Transformer XL| 8K | 2K | 50.3M | 10.94 |
> |Memorizing Transformer| 8K | 2K | 184.5M| 10.38 |
> |**MELODI S512+L256**| 8K | 2K | 23.1M | 10.20 |
> |**MELODI S768+L384**| 8K | 2K | 34.6M | **10.10** |
>
> *Challenges for Scaling MELODI Up*
>
> Scaling MELODI to larger context windows presents two primary challenges:
>
> 1. *Memory Capacity*: The memory requirements of MELODI grow with both the context window size and the batch sequence length. With our current hardware and codebase, we are unable to train MELODI with context windows exceeding 2K tokens. But this can be addressed by using TPUs with more memory and improving model parallelism in our codebase. We plan to investigate it in the future work.
>
> 2. *Backpropagation Through Time (BPTT)*: MELODI's recurrent connections necessitate BPTT, which further increases memory consumption. Each batch spans 8 context windows by default, allowing for BPTT across multiple windows. While crucial for the recurrent short-term memory, BPTT contributes to memory usage. As shown in the table above, using longer batch sequences (8K tokens) with 1K token windows yields better performance than shorter sequences (4K tokens) due to the increased BPTT depth.

---

> ### Author Response · Authors · 2024-11-20
> **Response from authors - part 2**
>
> $\color{blue}{\textbf{[Weakness 1.2]:}}$
>
> **Figure 5 could be highlighted, discussed and presented more clearly and explicitly in terms of the long context issue. It seems this experiment does give perplexity numbers for 192x64=12k token context windows, but it is a bit hidden amongst the rest of the experiments given the importance of these longer regimes.**
>
> Figure 5 provides insights into MELODI's ability to utilize information from a much larger effective context length than size of the context window. In this experiment, the context window size is fixed at 512 tokens, and the x-axis in Figure 5 represents the number of previous context windows covered by the long-term memory. Each window is compressed to 64 long-term tokens, e.g. an 8x compression ratio. For instance, the point labeled '192' indicates that the long-term memory stores information from the previous 192 context windows, so the effective context length is 192 windows × 512 tokens/window = 96K tokens.
> The results in Figure 5 show a clear performance improvement as the long-term memory coverage increases, especially from 4 to 32 previous windows.
>
> ---
> $\color{blue}{\textbf{[Weakness 2]:}}$
>
> **lack of fine-tuning experiments MELODI and the baselines. Evaluating the performance of fine-tuned MELODI models on downstream tasks would provide insights into its effectiveness in practical applications.**
>
> Thank you for the insightful suggestion. We totally agree that it is important to test MELODI on downstream tasks.  However, testing on downstream tasks will require larger models and more extensive pre-training than we currently have been able to do, so we are leaving those studies for future work.
>
> The goal of this paper was simply to determine whether the MELODI memory mechanism works, by using small models trained from scratch, and comparing those models against strong baselines which were also trained from scratch.
>
> In future work, we plan to explore techniques like LoRA (Low-Rank Adaptation) and/or fine-tuning to efficiently integrate MELODI's short-term and long-term memory mechanisms into pre-trained models. This will allow us to assess MELODI's effectiveness on various downstream tasks, such as summarization and question answering, and compare its performance against SOTA baselines.

---

> > ### Author Response · Authors · 2024-11-20
> > **Response from authors - part 3**
> >
> > $\color{blue}{\textbf{[Weakness 3 + Question 2]:}}$
> >
> > **Limited Analysis of Computational Efficiency More analysis and discussion of Memory Usage and Computational Time or computational complexity would be helpful. The paper could potentially benefit from discussions and comparisons of training and inference times for MELODI and the baselines. Could you provide a discussion and/or analysis of MELODI's computational efficiency, including complexity analysis or measurements and comparisons of training and inference times compared to baseline models? Latency is an issue in real world situations and it would be useful for the reader to understand the inference time implications here.**
> >
> > Thank you for the suggestion. To address your concern about computational cost, we provide an analysis of MELODI's computational complexity and report training and inference times compared to the baselines.
> >
> > *Computational Complexity*
> >
> > First, we define the notation used in our complexity analysis:
> >
> > $N$: Number of layers
> >
> > $W$: Number of context tokens per window
> >
> > $D$: Dimension of token embeddings
> >
> > $S$: Number of short-term memory tokens per window
> >
> > $Q$: Full length of long-term memory in the Memorizing Transformer
> >
> > $R$: MELODI's Long-term memory redunction rate compared to the Memorizing Transformer
> >
> > The table below presents a component-wise comparison of computational complexity between MELODI and the baseline models.
> >
> > | Method | Query-Key-Value | Self-Attention | Cross-Attention | Feedforward Network | Linear Token Mixing |
> > |---|:-:|:-:|:-:|:-:|:-:|
> > | Transformer XL | $O(NWD^2)$ | $O(NW^2D)$ | -- | $O(NWD^2)$ |--|
> > | Memorizing Transformer | $O(NWD^2)$ | $O(NW^2D)$ | $O(QWD)$| $O(NWD^2)$ | -- |
> > | **MELODI** | $O\left[N(W+S)D^2\right]$ | $O(NW^2D)$ | $O(\frac{QWD}{R})$ | $O[N(W+S)D^2]$ | $O\left[N(W+S)SD\right]$|
> >
> > As shown in the table, the Memorizing Transformer introduces additional complexity compared to TransformerXL due to the cross-attention computation over the long-term memory in its middle layer. MELODI reduces this cross-attention complexity by a factor of $R$ compared to the Memorizing Transformer. However, it introduces additional computations in the query-key-value, feedforward network, and linear token mixing components. The extent of these additional computations is controlled by the number of short-term memory tokens, $S$, which is about a quarter of window length $W$.
> >
> > *Training and Inference Times*
> >
> > To assess MELODI's computational efficiency, we measured its training and inference time and compared it against the baselines.  All experiments were conducted using 12-layer networks with an embedding dimension of 1024 on TPU v6e. We evaluated performance with batch sequence lengths of 4K and 8K tokens and window lengths of 512 and 1K tokens. The results are presented in the table below.
> >
> > | Method | Sequence Length | Window Length | Training time (sec) | Inference time (sec) |
> > |---|:-:|:-:|:-:|:-:|
> > | Transformer XL | 4K | 512 | 0.052 | 0.022 |
> > | Memorizing Transformer | 4K | 512 | 0.094 | 0.033 |
> > | **MELODI S128+L64** | 4K | 512 | 0.076 | 0.023 |
> > | **MELODI S192+L96** | 4K | 512 | 0.097 | 0.027 |
> > ||
> > | Transformer XL | 4K | 1K | 0.068 | 0.027 |
> > | Memorizing Transformer | 4K | 1K | 0.108 | 0.036 |
> > | **MELODI S256+L128** | 4K | 1K | 0.094 | 0.028 |
> > | **MELODI S384+L192** | 4K | 1K | 0.111 | 0.031 |
> > ||
> > | Transformer XL | 8K | 1K | 0.108 | 0.045 |
> > | Memorizing Transformer | 8K | 1K | 0.196 | 0.067 |
> > | **MELODI S256+L128** | 8K | 1K | 0.164 | 0.047 |
> > | **MELODI S384+L192** | 8K | 1K | 0.204 | 0.054 |
> >
> > As shown in the table, MELODI consistently demonstrates faster inference times than the Memorizing Transformer. Impressively, the MELODI S128+L64 variant approaches the inference speed of TransformerXL (e.g., 0.022 vs. 0.023 seconds with an 4K sequence length and 512 window length).
> >
> > However, when considering training time, both MELODI variants are comparable to the Memorizing Transformer but slower than TransformerXL. This suggests that the additional computations introduced by MELODI's short-term memory component (in the query-key-value, feedforward network, and linear token mixing) are relatively inexpensive during inference (forward pass) but become more costly during backpropagation.

---

> > > ### Author Response · Authors · 2024-11-20
> > > **Response from authors - part 4**
> > >
> > > $\color{blue}{\textbf{[Weakness 4 + Question 3]:}}$
> > >
> > > **Experiments are almost exclusively performed providing test set perplexity results. While there are many downstream tasks that could benefit from models yielding better perplexity, it would be more compelling to see the impact on other kinds of tasks and metrics where long range information is important. The paper highlights the potential of MELODI for various natural language processing applications, such as document summarization, question answering, and machine translation. Have you considered conducting preliminary experiments on any of these downstream tasks to demonstrate the practical benefits of MELODI in specific applications? With those kinds of experiments and answers to the questions above I could potentially score this paper higher.**
> > >
> > > We agree that evaluating MELODI on downstream tasks like summarization and question answering is an important next step. This will involve scaling up the model and incorporating MELODI into a pre-trained LLMs using fine-tuning and/or LoRA, which we plan to explore in future work.
> > >
> > > *New Task: Masked Next Token Prediciton*
> > >
> > > Although a comprehensive evaluation on downstream tasks is impractical within the rebuttal period, we introduce a novel task designed to further evaluate MELODI's capacity for handling long-range dependencies.
> > >
> > > Similar to our primary evaluation, we use next token prediction and measure performance with perplexity. However, we introduce a key challenge: masking out a portion of the input tokens. Unlike BERT's bidirectional masking approach, this new task employs a unidirectional, left-to-right prediction scheme.
> > >
> > > *Why evaluate on this task?*
> > >
> > > This masking strategy presents a unique challenge to the model. By disrupting the local context with masked tokens, we force the model to rely on information from more distant, unmasked tokens to accurately predict the next word. This effectively evaluates the model's ability to capture and utilize longer range dependencies. A model with strong long-range capabilities will demonstrate lower perplexity on this task, indicating its proficiency in integrating information across extended sequences.
> > >
> > > *Experimental Results*
> > >
> > > To evaluate MELODI's performance on this new task, we conducted experiments using the PG-19 dataset (with the T5 vocabulary) and two different architectures: 12-layer and 36-layer. We applied random masking with three masking ratios (0.125, 0.25, and 0.5) to systematically vary the degree of contextual disruption.
> > >
> > > As expected, perplexity increased across all models as the masking ratio increased, indicating the growing difficulty of the task. However, MELODI consistently outperformed both TransformerXL and the Memorizing Transformer across all masking ratios and architectural configurations. These results, presented in the table below, strongly suggest that MELODI exhibits superior capabilities in capturing and utilizing long-range dependencies compared to these baseline models.
> > >
> > > | Method | Mask Ratio | Num of Layers |Perplexity $\downarrow$ | | Num of Layers |Perplexity $\downarrow$ |
> > > |---|:-:|:-:|:-:|---|:-:|---|
> > > |Transformer XL| no mask | 12 | 11.54 | | 36 | 9.61 |
> > > |Memorizing Transformer| no mask | 12 | 10.74 | | 36 | 8.92 |
> > > |**MELODI S128+L64**| no mask | 12 | 10.61 | | 36 | 8.81 |
> > > |**MELODI S192+L96**| no mask | 12 | **10.48** || 36 | **8.70** |
> > > ||
> > > |Transformer XL| 0.125 | 12 | 16.08 | | 36 | 13.10 |
> > > |Memorizing Transformer| 0.125 | 12 | 14.91 | | 36 | 12.15 |
> > > |**MELODI S128+L64**| 0.125 | 12 | 14.58 | | 36 | 11.97 |
> > > |**MELODI S192+L96**| 0.125 | 12 | **14.39** || 36 | **11.74** |
> > > ||
> > > |Transformer XL| 0.25 | 12 | 22.50 | | 36 | 18.18 |
> > > |Memorizing Transformer| 0.25 | 12 | 20.76 | | 36 | 16.64 |
> > > |**MELODI S128+L64**| 0.25 | 12 | 20.09 | | 36 | 16.21 |
> > > |**MELODI S192+L96**| 0.25 | 12 | **20.00** || 36 | **15.98** |
> > > ||
> > > |Transformer XL| 0.5 | 12 | 65.51 | | 36 | 51.23 |
> > > |Memorizing Transformer| 0.5 | 12 | 56.28 | | 36 | 46.80 |
> > > |**MELODI S128+L64**| 0.5 | 12 | 53.63 | | 36 | 45.57 |
> > > |**MELODI S192+L96**| 0.5 | 12 | **52.59** || 36 | **44.05** |

---

> > ### Comment · Reviewer_XNtf · 2024-11-25
> > **I have read the author response and I maintain my relatively strong accept rating.**
> >
> > Thanks for the additional experiments, they may help the paper have greater impact.

---

> > > ### Author Response · Authors · 2024-11-25
> > > **Appreciation for your support**
> > >
> > > Thank you for taking the time to read our response. We appreciate your positive feedback and believe the additional experiments will indeed contribute to the paper's impact. We are grateful for your thoughtful review and support of our work.

---

> ### Author Response · Authors · 2024-11-20
> **Response from authors - part 5**
>
> *Slower Perplexity Increase for MELODI with Increasing Mask Ratio*
>
> Interestingly, even with a high masking ratio of 0.5, the 12-layer MELODI variant (S192+L96) achieves a perplexity of 52.59, which is remarkably close to the 36-layer TransformerXL's perplexity of 51.23. This suggests that MELODI can achieve comparable performance with significantly fewer parameters.
>
> To further analyze this trend, we examined the change in perplexity as the masking ratio increases: 0 → 0.125, 0.125 → 0.25, and 0.25 → 0.5. As shown in the table below, the perplexity increase for MELODI is consistently smaller than that observed for the baseline models. This indicates that MELODI is less affected by the increasing disruption of local context, further highlighting its superior ability to capture and leverage long-range dependencies.
>
> | Method | Num of Layers | Mask Ratio 0 $\rightarrow$ 0.125 | Mask Ratio 0.125 $\rightarrow$ 0.25 | Mask Ratio 0.25 $\rightarrow$ 0.5 |
> |---|:-:|:-:|:-:|:-:|
> | Transformer XL | 12 | +4.54 | +6.42 | +43.01 |
> | Memorizing Transformer | 12 | +4.17 | +5.85 | +35.52 |
> | **MELODI S128+L64** | 12 | +3.97 | +5.51 | +33.54 |
> | **MELODI S192+L96** | 12 | +3.91 | +5.61 | +32.59 |
> ||
> | Transformer XL | 36 | +3.49 | +5.08 | +33.05 |
> | Memorizing Transformer | 36 | +3.23 | +4.49 | +30.16 |
> | **MELODI S128+L64** | 36 | +3.16 | +4.24 | +29.36 |
> | **MELODI S192+L96** | 36 | +3.04 | +4.24 | +28.07 |

---

> > ### Author Response · Authors · 2024-11-22
> > **Reminder: Author Response and Follow Up**
> >
> > Dear Reviewer XNtf,
> >
> > We hope this message finds you well. We have posted our responses to your valuable feedback and appreciate you taking the time to review them.  We understand the demands on your time and deeply appreciate your active participation in this discussion.
> >
> > As the author-reviewer discussion period nears its close, we want to reiterate that we are available to answer any further questions or concerns you may have about our rebuttal. Your continued insights are crucial in helping us further refine our manuscript.
> >
> > Please don't hesitate to reach out if you have any further thoughts or inquiries.
> >
> > Sincerely,
> >
> > The Authors

---

### Official Review · Reviewer_9LJX · 2024-11-04

**Soundness:** 2
**Presentation:** 2
**Contribution:** 3
**Rating:** 5
**Confidence:** 2

**Summary:**

This paper proposes a new context compression neural network for the long context in today's LLMs. With a hierarchical structure, it achieved a decent performance by reducing the memory footprint by a factor of 8.

**Strengths:**

1. The methodology seems novel and makes a lot of sense in dividing the tasks into long term and short term memory in the compression.
2. The empirical results seem promising.

**Weaknesses:**

1. Without prior knowledge, it's not clear that how the memory network works with the LLMs and this is also not reported as preliminary knowledge.
2. The results only show the efficiency part and do not show the effect of memory network on the performance of LLM.
3. Codes are not shared for checking the implementation.

**Questions:**

1. A preliminary session about the history of recent advances about memory compression would be much appreciated.

---

> ### Author Response · Authors · 2024-11-20
> **Response from authors - part 1**
>
> We sincerely thank the reviewer for the thoughtful feedback and valuable suggestions, which have significantly improved the quality of our paper.
>
> $\color{blue}{\textbf{[Weakness 1 + Question 1]:}}$
>
> **Prior knowledge about how the memory network works with the LLMs. Adding a preliminary session about the history of recent advances about memory compression.**
>
> Thank you for the suggestion. Although the prior works are discussed in the related work section (i.e. Section 4 after the experiment section), we agree that providing the context on memory networks and compression earlier in the paper would improve readability. We will move the related work section before the method section in the revised manuscript.
>
> For your convenience, we have included an overview of relevant work on memory networks and compression below:
>
> *Memory in language models:*
>
> Long Short-Term Memory (LSTMs) [1] use token-level recurrence to compress prior context into a state vector, which is a limited form of memory.  With the advent of Transformers [2], the focus has shifted to memory mechanisms operating at the level of the context window; this shift allows blocks of tokens (i.e. all tokens within the window) to be processed in parallel. The Block Recurrent Transformer [4] and Recurrent Memory Transformer (RMT) [5] integrate recurrent mechanisms inspired by LSTMs into the Transformer architecture, but the recurrence is over blocks, rather than individual tokens.
>
> Transformer-XL [3] introduces a caching mechanism to store key-value (KV) pairs from the preceding context window as a form of short-term memory.  Memorizing Transformer [9] dramatically expands the size of the cache to store KV pairs for long-term memory, but only uses the long-range cache in a single layer, for efficiency reasons. MemoryLLM [10] incorporates long-term memory in every layer, incurring a substantial memory overhead. Infini-Transformer [6] explore the use of additional memory like Hopfield Networks [7, 8]. LONGMEM [11] improves over Memorizing Transformer by introducing a SideNet for memory retrieval and fusion. You Only Cache Once (YOCO) [12] shows that long-term KV cache is reuable for the latter half of the network, significantly improving pre-filling efficiency by enabling early exit.
>
> MELODI, in contrast, integrates integrates both short-term and long-term memory into a transformer model via compression.
>
> [1] Sepp Hochreiter and Jurgen Schmidhuber, "Long short-term memory", Neural Computation, 9(8): 1735–1780, 1997.
>
> [2] Ashish Vaswani, et al, "Attention is all you need", Advances in Neural Information Processing Systems, 2017.
>
> [3] Zihang Dai, et al, "Transformer-XL: Attentive language models beyond a fixed-length context", Annual Meeting of the Association for Computational Linguistics, 2019.
>
> [4] DeLesley Hutchins, et al, "Block-recurrent transformers", Advances in Neural Information Processing Systems, 2022.
>
> [5] Aydar Bulatov, et al, "Recurrent memory transformer", Advances in
> Neural Information Processing Systems, 2022.
>
> [6] Tsendsuren Munkhdalai, et al, "Leave no context behind: Efficient
> infinite context transformers with infini-attention", 2024. https://arxiv.org/abs/2404.07143.
>
> [7] J. J. Hopfield. "Neural networks and physical systems with emergent collective computational abilities", Proceedings of the National Academy of Sciences of the United States of America, 79(8): 2554–2558, April 1982.
>
> [8] Hubert Ramsauer, et al, "Hopfield networks is all you need", 2021. https://arxiv.org/abs/2008.02217.
>
> [9] Yuhuai Wu, et al, "Memorizing transformers", International Conference on Learning Representations, 2022.
>
> [10] Yu Wang, et al, "Memoryllm: Towards self-updatable large language models", 2024. https://arxiv.org/abs/2402.04624.
>
> [11] Wang, Weizhi, et al. "Augmenting language models with long-term memory." Advances in Neural Information Processing Systems 36 (2024).
>
> [12] You Only Cache Once: Decoder-Decoder Architectures for Language Models, 2024.

---

> ### Author Response · Authors · 2024-11-20
> **Response from authors - part 2**
>
> *Compression:*
>
> Recent work has explored using summary tokens for compression in Transformers [5, 13, 14, 15]. Recurrent Memory Transformer (RMT) [5] utilizes the output of summary tokens recurrently as short-term memory. AutoCompressor [14] aggregates summary tokens across segments to generate a summary representation for long documents used in retrieval tasks. Gisting [16] applies this technique to compress long prompts. The In-context Autoencoder (ICAE) [15] further incorporates LoRA fine-tuning [17] for context compression, while TransformerFAM [18] introduces feedback attention to enhance performance.
>
> Unlike these methods that compress input tokens, MELODI compresses network activations over multiple layers.
>
> [13] Jack W Rae, et al, "Compressive transformers for long-range sequence modelling", arXiv preprint, 2019. https://arxiv.org/abs/1911.05507.
>
> [14] Alexis Chevalier, et al, "Adapting language models to compress contexts", Proceedings of the 2023 Conference on Empirical Methods in Natural Language Processing, 2023.
>
> [15] Tao Ge, et al, "In-context autoencoder for context compression in a large language model", 2024. https://arxiv.org/abs/2307.06945.
>
> [16] Jesse Mu, et al, "Learning to compress prompts with gist tokens", 2024.
> https://arxiv.org/abs/2304.08467.
>
> [17] Edward J. Hu, et al. "Lora: Low-rank adaptation of large language models", 2021. https://arxiv.org/abs/2106.09685.
>
> [18] Dongseong Hwang, et al. "Transformerfam: Feedback attention is working memory", 2024. URL https://arxiv.org/abs/2404.09173.
>
> ---
> $\color{blue}{\textbf{[Weakness 2]:}}$
>
> **The results only show the effiency part but not the effect of memory on the performance of LLM.**
>
> We apologize for the lack of clarity. Our results demonstrate both the *efficiency* and *impact on LLM performance* of MELODI's memory mechanisms.
>
> * ***Impact of memory on LLM performance:*** To showcase the impact of memory on LLM performance, we compare MELODI to Transformer XL, a transformer-based language model with a KV cache of the previous context window.  MELODI uses about the same amount of memory as TransformerXL, but significantly outperforms Transformer XL in terms of perplexity. For instance in Table 3 (of the paper), on the PG-19 dataset with the T5 vocabulary, *MELODI S192+L32* achieves a perplexity of 10.51, compared to 11.41 for Transformer XL, demonstrating a clear improvement. This highlights the effectiveness of MELODI's hierarchical memory mechanism in capturing and utilizing long-range context.
>
> * ***Efficiency:*** We also compare MELODI to the Memorizing Transformer. MELODI achieves comparable or slightly better perplexity as memorizing transformer, but uses approximately 8 times less memory.
>
> ---
> $\color{blue}{\textbf{[Weakness 3]:}}$
>
> **Codes are not shared for checking the implementation.**
>
> Thank you for your feedback. We understand the importance of code sharing for reproducibility. Our organization's policy requires an internal review before code release, which can take some time. We are committed to making our code publicly available upon the paper's acceptance. In the meantime, we are happy to answer any specific questions you may have about the implementation.

---

> > ### Author Response · Authors · 2024-11-22
> > **Reminder: Author Response and Follow Up**
> >
> > Dear Reviewer 9LJX,
> >
> > We hope this message finds you well. We have posted our responses to your valuable feedback and appreciate you taking the time to review them.  We understand the demands on your time and deeply appreciate your active participation in this discussion.
> >
> > As the author-reviewer discussion period nears its close, we want to reiterate that we are available to answer any further questions or concerns you may have about our rebuttal. Your continued insights are crucial in helping us further refine our manuscript.
> >
> > Please don't hesitate to reach out if you have any further thoughts or inquiries.
> >
> > Sincerely,
> >
> > The Authors

---

> > > ### Author Response · Authors · 2024-11-25
> > > **Re: Author Response and Follow Up**
> > >
> > > Dear Reviewer 9LJX,
> > >
> > > We wanted to follow up on our previous message and see if you had any further thoughts or questions about our response to your valuable feedback.  We understand how busy you are, and we truly appreciate you taking the time to engage with our work.
> > >
> > > The author-reviewer discussion period is closing in two days. We are very grateful for your insights and would be happy to clarify any aspects of our rebuttal or address any remaining concerns.
> > >
> > > Please don't hesitate to reach out if you have any further questions.
> > >
> > > Sincerely,
> > >
> > > The Authors

---

> > > > ### Author Response · Authors · 2024-11-27
> > > > **Revised Manuscript with Incorporated Feedback**
> > > >
> > > > Dear Reviewer 9LJX,
> > > >
> > > > We hope this email finds you well.
> > > >
> > > > Following up on our previous message, we wanted to let you know that we have carefully revised our manuscript based on your valuable feedback. We have incorporated your suggestions as follows:
> > > >
> > > > * ***Prior knowledge about the memory network and compression:*** see Line 86-120 in Section 2.
> > > >
> > > > * ***Results demonstrate both the efficiency and impact on LLM performance of MELODI's memory mechanisms:*** see Line 417-429 in Section 4.3
> > > >
> > > > The revised manuscript is available for your review. We believe that these changes strengthened the paper and addressed your concerns.
> > > >
> > > > The author-reviewer discussion period is closing soon. If you have any further questions or require any clarifications, please don't hesitate to reach out. We appreciate your time and consideration.
> > > >
> > > > Sincerely,
> > > >
> > > > Authors

---

> > > > > ### Author Response · Authors · 2024-12-02
> > > > > **Appreciating Your Feedback - Discussion Closing Tomorrow**
> > > > >
> > > > > Dear Reviewer 9LJX,
> > > > >
> > > > > As the author-reviewer discussion period comes to a close tomorrow, we wanted to gently remind you that we've revised our manuscript based on your feedback and would welcome any final thoughts you might have.
> > > > >
> > > > > Thank you again for your time and contributions to our work.
> > > > >
> > > > > Sincerely,
> > > > >
> > > > > Authors

---

### Author Response · Authors · 2024-11-27
**Revised Manuscript**

We thank all reviewers for the thoughtful feedback and valuable suggestions. We have carefully incorporated the valuable suggestions, comments, and discussions provided by all reviewers. The manuscript has been updated accordingly.

For your convenience, we have compiled a list of all changes, along with their corresponding line numbers.  All modifications are highlighted in $\color{blue}{blue}$ within the revised manuscript (in both main paper and appendix).

| Description | Location in the revised manuscript | Reviewer | Related Weakness/Question |
|:--|:--|:-:|:--|
|Moving the related work section before the method section| Line 86-120 in Section 2 | 9LJX | Weakness 1, Question 1 |
|Results demonstrate both *efficiency* and *impact on LLM performance* of MELODI's memory | Line 417-429 in Section 4.3 |9LJX | Weakness 2|
||
|Scaling up context length| Line 952-1014 + Table 7 in Appendix D.3  | XNtf | Weakness 1, Question 1 |
|Clarification of Figure 5 | Line 443-457 in Figure 5's caption | XNtf | Weakness 1 |
|Limitations and future work | Line 756-782 in Appendix A | XNtf | Weakness 2 |
| Analysis of computational complexity and training/inference times | Line 1169-1223 + Table 14, 15 in Appendix E.3 and E.4 | XNtf | Weakness 3, Question 2 |
| New task: masked next token prediction | Line 1048-1079 + Table 9 in Appendix D.5 | XNtf | Weakness 4, Question 3 |
||
|New task: masked next token prediction| Line 1048-1079 + Table 9 in Appendix D.5  | hBrU | Weakness 1 |
|Discussion of efficient parallel training| Line 827-847 in Appendix C.1 | hBrU | Weakness 2 |
|Chunk-wise processing in generation| Line 848-859 in Appendix C.2 | hBrU | Weakness 3 |
|Scaling up model size| Line 915-950 + Table 6 in Appendix D.2 | hBrU | Weakness 4 |
|Comparison with Transformer using longer context | Line 1016-1046 + Table 8 in Appendix D.4 | hBrU | Question 1 |
|Comparison with YOCO | Line 101 in Section 2 (Related Work), Line 806-824 in Appendix B | hBrU | Question 2 |
|Limitations and future work | Line 756-782 in Appendix A | hBrU | Followup |
||
|Comparison with LongMem | Line 100 in Section 2 (Related Work), Line 791-805 in Appendix B | i9z9 | Weakness 1, Question 1 |
|Ablation on long-term layer position $M$ | Line 1105-1108 + Table 11 in Appendix D.6 | i9z9 | Weakness 2, Question 2 |
|Memory measured as the number of floats | Line 311-317 in Section 3.3 | i9z9 | Weakness 2, Question 2 |
|Experiments with more long-term memory layers in Figure 8| Line 517-523 + updated Figure 8 in Section 4.4 | i9z9 | Weakness 3, Question 3 |
|Memory Size vs. Model Size | Line 1127-1167 + Table 12, 13 in Appendix E.2 | i9z9 | Followup |

Thanks again for the constructive efforts in the comments and reviews.

Authors

---

### Meta-Review · Area_Chair_Xiyy · 2024-12-20

**Metareview:**

This paper proposes a novel and slightly intricate architecture for modelling long context. It is based on two types of layers, short term and long term. The short term modules treat a short span of context and introduces something akin to a recurrence with summary tokens. The long term module cross attends to memory tokens from previous short term modules. The proposed architecture is mostly evaluated on language modeling tasks. The reviewers have raised some practical concerns, and pointed out some lacking baselines (the ones in the paper were old), slightly limited scale of experiments, as well as lack of specialized evaluations (long-context specific, or needle in a haystack). Nonetheless, given the interesting methodological contributions in this work, I recommend accepting this paper for ICLR 2025.

**Additional Comments On Reviewer Discussion:**

Most of the concerns raised by the reviewers have been addressed in the rebuttal and the reviewers acknowledged that in the discussion. While some small weaknesses remained, I think that the positives clearly outweigh the small downsides.

---

### Decision · Program_Chairs · 2025-01-22

Accept (Poster)